# Integrated biosensor platform based on graphene transistor arrays for real-time high-accuracy ion sensing

Mantian Xue [1,4] ✉, Charles Mackin [2,4], Wei-Hung Weng[1], Jiadi Zhu[1], Yiyue Luo[1], Shao-Xiong Lennon Luo [3], Ang-Yu Lu [1], Marek Hempel[1], Elaine McVay[1], Jing Kong [1] & Tomás Palacios [1] ✉

Two-dimensional materials such as graphene have shown great promise as biosensors, but suffer from large device-to-device variation due to non-uniform material synthesis and device fabrication technologies. Here, we develop a robust bioelectronic sensing platform composed of more than 200 integrated sensing units, custom-built high-speed readout electronics, and machine learning inference that overcomes these challenges to achieve rapid, portable, and reliable measurements. The platform demonstrates reconfigurable multi-ion electrolyte sensing capability and provides highly sensitive, reversible, and real-time response for potassium, sodium, and calcium ions in complex solutions despite variations in device performance. A calibration method leveraging the sensor redundancy and device-to-device variation is also proposed, while a machine learning model trained with multi-dimensional information collected through the multiplexed sensor array is used to enhance the sensing system's functionality and accuracy in ion classification.

Smart sensors, such as sweat sensors, targeting various physiologically-relevant biomarkers in biofluids have recently demonstrated great potential for health-tracking and medical diagnosis[1–3]. Such systems are commonly multiplexed to identify different analytes or provide active calibration[1]. Two-dimensional (2D) materials are particularly promising in biochemical sensing applications thanks to their large-surface-to-volume ratio, which allows the sensor channel to be readily modulated upon chemical changes near the surface, translating chemical signals into the electrical or optical domain with enhanced sensitivity. Graphene is the most widely studied material among all 2D materials and provides the most mature material synthesis[4]. It has the largest surface-to-volume ratio[5] and exhibits a number of promising characteristics. The high carrier mobility[6], which translates into high transconductance, makes graphene a more desirable transducer compared to organic materials such as poly(3,4-ethylenedioxythiophene) polystyrene sulfonate (PEDOT:PSS). In contrast to traditional metal oxide-based chemo-resistor sensors, which

suffer from high humidity sensitivity, graphene is stable and chemically inert[7]. Thanks to its mechanical flexibility[8], graphene can potentially enable innately flexible form-factors in contrast to silicon-based sensing systems. Graphene is also compatible with a large variety of surface functionalization chemistries, making it a promising sensing material for large-area arrays of multiplexed sensors[9].

Despite advances in material synthesis, large-area integration of devices based on graphene and other novel materials still suffer from strong device-to-device variability caused by intrinsic defects[10], gate oxide nonuniformities[11], and parasitic effects[12]. Device fabrication also introduces additional variability in sensor response from batch to batch[10]. Most sensor papers previously reported contain a single sensor for each type of analyte, where the performance of each sensor is evaluated and optimized separately[1,3,13,14]. This brings into question the reproducibility and reliability of such devices when applied to real-world applications, including complex physiological samples.

[1]Department of Electrical Engineering & Computer Science, Massachusetts Institute of Technology, Cambridge, MA, USA. [2]IBM Research–Almaden, 650 Harry Road, San Jose, CA, USA. [3]Department of Chemistry and Institute for Soldier Nanotechnologies, Massachusetts Institute of Technology, Cambridge, MA, USA. [4]These authors contributed equally: Mantian Xue, Charles Mackin. ✉e-mail: mxue@mit.edu; tpalacios@mit.edu

Here we demonstrate an approach to overcome the challenges in 2D material-based sensing devices and achieve high performance and enhanced functionality. Rather than focusing on the improvement of intrinsic material quality, fabrication uniformity, or surface functionalization, we develop a high-density graphene-based sensor array platform to overcome the large degree of variability of advanced materials. We fabricate arrays (16×16) of graphene devices to provide more than 200 working sensing units for each chip, and configure them into multi-ion sensors by functionalizing the surface with three different ion-selective membranes (ISMs). Ionized calcium, potassium, and sodium were chosen as analytes of interest due to their commonplace in diagnostic tests and their physiological importance in blood, urine, and sweat[3,15,16]. They help regulate fluid balance, muscle contractions, nerve system transmissions, and glandular secretion. These ions are essential in evaluating human physiological status, as they are indicators for diuretic use, gastrointestinal losses, kidney disease, parathyroid condition, thyroid disease, cardiac failure etc[17–19]. Attempts have been made with graphene-based ion sensors in the past, but such sensors mostly target only one type of ion using a very limited number of test devices[20–22]. The reproducibility of the sensor performance from device-to-device, wafer-to-wafer, and lot-to-lot is mostly unreported. We demonstrate near-ideal sensitivity, excellent reversibility, and large detection range for each type of sensor despite non-uniformity in individual devices. The variations and imperfections in material synthesis and device fabrication can be leveraged by statistical analysis and machine learning algorithms. A profile-matching calibration method utilizing sensor non-uniformity and redundancy is introduced to eliminate the need for multiple calibration solutions,

which is especially useful for sensing applications targeting portability and field use. A Random Forest algorithm is used to quantify analyte concentrations in the presence of multiple-ions. The abundance ($N > 200$) and multiplexity of sensors and sensor types are shown beneficial for improving model accuracy. We demonstrate that system-level co-design of sensing arrays and algorithms significantly improves sensor performance thus enabling rapid prototyping and in-depth data analysis in spite of the limitations present in graphene and other advanced 2D materials.

## Results

### ISM functionalized graphene sensor array

The graphene sensing chip used in this study is fabricated on a 4-inch, 200 μm thick glass wafer (Supplementary Fig. 1b). Each sensing unit consists of a $30 \times 30$ μm graphene channel with two Ti (5 nm)/Au (150 nm) source/drain electrodes. The optical image of an as-fabricated graphene sensor array is shown in Supplementary Fig. 1c. The quality of the intrinsic graphene film is analyzed by Raman Spectroscopy as shown in Supplementary Fig. 2a, b. The weak *D* band indicates minimal defects in the graphene film and the excellent *2D/G* ratio mapping shows the sheet is single layer. A 500 nm SU-8 passivation film was spin-coated on top of the devices and patterned to leave openings in the sensing areas above the graphene channels. A material jetting printer is utilized to deposit various functionalization chemistries onto the sensing area with precise lateral control as illustrated in Fig. 1a. A measurement system consisting of a custom-built printed-circuit board (PCB) and microcontroller allows for rapid acquisition of high-quality data from a large number of sensors in a convenient

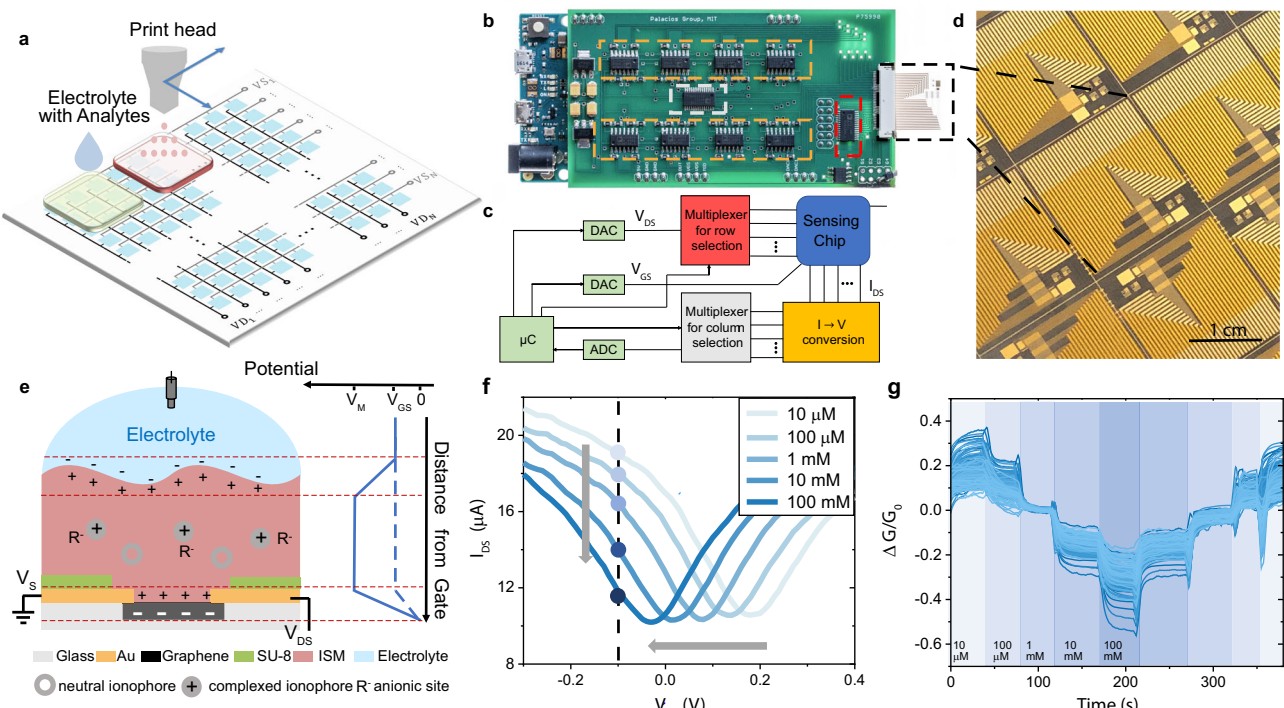

**Fig. 1 | Highly-integrated sensing system. a** Schematic of the sensing chip consisting of $N \times N$ sensor units ($N = 16$ in this paper). Different surface functionalization membranes can be printed onto different regions of the sensor array, allowing multiplexed measurements. **b** Optical photograph of the measurement system and the sensor array. **c** Block diagram of the measurement system color-coded with the dashed boxes in **b**. **d** Microscope image of the graphene sensing arrays on a glass wafer. Scale bar: 1 cm. **e** Schematic of the individual sensing unit with ion-sensitive surface functionalization membrane. The electrostatic potential as a function of

distance from graphene surface is shown on the right. ISM: ion sensitive membrane; $V_M$: membrane potential; $V_{GS}$: gate to source voltage; $V_{DS}$: drain to source voltage; $V_S$: potential at source. **f** Leftward shifts of *I–V* curves observed in a typical device from a Na$^+$ ISM functionalized sensing chip with increased sodium ion concentration and $V_{DS} = 300$ mV. Black dashed line indicates the current level at $V_{GS} = -0.1$ V and a decrease in current is observed due to the left-shift of the *I–V* curves. **g** Normalized conductance transient responses of 215 working sensing units to changing concentrations in ionized sodium at $V_{DS} = 300$ mV, and $V_{GS} = -0.1$ V.

manner (Fig. 1b, c). The measurement system also houses an external low-profile Ag/AgCl reference electrode to provide a stable reference potential for data acquisition. When performing liquid-phase measurements, the sensor array chip is dipped into a beaker with 10 ml of testing solution at room temperature along with the low-profile reference electrode, which functions as a global top-gate for the sensing chip. The change in the source-drain current $I_{DS}$ for each row and column combination of the sensor array are automatically measured as a function of gate-source and drain-source voltages, $V_{GS}$ and $V_{DS}$. For a standard $V_{GS}$ sweep (−0.6 to 0.9 V with 20 mV/s sweeping speed), it takes roughly three minutes to finish measuring all 256 devices. The drain current with respect to gate voltage (will be referred to as "$I–V$" characteristics) as well as the drain current with respect to drain voltage characteristics of the sensor chip before functionalization are shown in Supplementary Fig. 3a, b. Non-functional pixels were filtered out using the criteria outlined in Supplementary Note 2 and the average yield for the sensing chip is >80%. Large device-to-device variations in terms of current level, channel resistance, Dirac Point (the location of the minimum conduction point) and shape of the $I–V$ characteristics are present for the >200 working devices on one sensing chip (Supplementary Fig. 3). A compact electrolyte-gated graphene-based field-effect transistor model was then applied to extract parameters like contact resistance, carrier concentration and mobility[23,24] to further showcase the variation within one sensing chip (Supplementary Fig. 7). Such variations result from a combination of material and fabrication non-uniformities. In the following section, we will demonstrate how to overcome and take advantage of such significant device-to-device variation.

The versatility of the sensing system is demonstrated by characterizing the performance of the sensors using different ISMs. ISMs in this work make use of charge neutral ionophores and draw inspiration from biological cell membranes. Ionophores are lipophilic molecules that can selectively bind to an ion of interest[25], which provides sensitivity and selectivity towards targeting ions along with pH independency[26,27]. Ionophores are assumed confined to the membrane phase due to their lipophilic properties. When a neutral ionophore is used, lipophilic ion sites with charge opposite to the analyte ion−in this case anionic sites−have to be added in order to suppress the extraction of chloride into the membrane[28–30]. Previous research shows that the concentration of these ionic sites within the membrane can be optimized to effectively reduce the response time, lower the electrical resistance of the membrane and increase the selectivity[28].

The operating principle of the sensor is based on the channel modulation of the graphene electrolyte-gated field-effect transistor (EGFET)[23,24] as the cation diffuses into the membrane. The transport of ions through the interface between the electrolyte and the ISM is governed by the Nernst equation[29] (detailed derivation is provided in the Supplementary Note 1). In equilibrium, diffusion of cations across the membrane is counterbalanced by the electric field induced by the ions. This potential is depicted in Fig. 1e at the electrolyte-membrane interface. As the concentration of the cations in the solution is increased, the electric field required to counterbalance diffusion must increase. Thus, the potential increases with increasing ion concentration. Because the polarity of the potential is aligned with the polarity of the source-gate voltage $V_{GS}$, an additional potential drop is added on the graphene channel as shown in Fig. 1e. In the presence of higher ion concentrations, a further accumulation of electrons occurs in the graphene channel and a larger electron current (lower hole current) is achieved for the same $V_{GS}$. Hence, increasing cation concentration results in a more n-doped channel and thus a leftward shift of the graphene $I–V$ characteristic. Figure 1f is a representative $I–V$ characteristic for a Na$^+$ ISM functionalized sensor measured in electrolytes containing various concentrations of sodium ions at ambient room temperature. The temperature sensitivity of graphene is negligible for ion sensing as shown in Supplementary Fig. 4. A Nernstian leftward shift of the $I–V$

characteristic was observed for increases in sodium ion concentration. Full $I–V$ curves were measured in each solution multiple times with the same drain voltage and gate voltage range. Hysteresis in graphene's $I–V$ characteristics is mitigated by applying a slow voltage sweeping rate of 20 mV/s. We obtain stable and reproducible measurements for bare and functionalized graphene despite the considerable variation among devices (Supplementary Fig. 5).

The sensor chip can also be configured to measure the transient response. In this case, both gate and source-drain voltage are held constant, while the chip is immersed in different solutions. The change in channel conductance shown in Fig. 1g is a typical transient response for a Na$^+$-ISM-functionalized chip with 215 working devices tested in dilutions of ionized sodium spanning several orders of magnitude. The changes in device conductance are normalized with respect to their responses in 1 mM NaCl solution. Spikes in data represent transition times during which the sensors were transferred from one solution to the next. When the gate is biased at the hole-conducting region (left side of $I–V$ curves), the conductance of the channel decreases with higher ion concentration due to the Nernstian leftward shift for cations as indicated by the black dotted line on Fig. 1f. Similar transient responses were also obtained for potassium and calcium ISMs.

The sensitivities of the functionalized sensors were extracted from the $I–V$ characteristics from three chips functionalized with K$^+$, Na$^+$, and Ca$^{2+}$ ISMs. The shift of the $I–V$ curve is quantified by tracking the Dirac Point, which was estimated by polynomial fitting the discretized $I–V$ characteristics and finding the minimum. Further details are provided in the Supplementary Note 3 and Supplementary Fig. 6. The average sensitivity plots at room temperature for K$^+$ ISM, Na$^+$ ISM, and Ca$^{2+}$ ISM functionalized sensor chips are depicted in Fig. 2a–c. The histogram plots fitted with Gaussian distributions show the variation of individual sensitivities for each type of sensor. All characterizations of ISM functionalized sensors were completed at least twice with different batches of sensor chips with results remaining consistent (Supplementary Fig. 9). Prior to averaging, the large variation in device behavior evidenced by the $I–V$ characteristics (Supplementary Fig. 3 and Supplementary Fig. 7) along with potential variations in membrane quality led to non-uniformity in ion sensitivities. Some devices show super-Nernstian behavior, which can be explained by uncontrolled charge transfer due to defect sites on graphene induced by fabrication process (Supplementary Note 1). By obtaining a large amount of data from all devices on the chip, one can overcome the noise signal that arises from device-to-device variation and capture the universal trend and enhance sensor signals by averaging. After averaging over the extracted Dirac Point, all three chips exhibit excellent Nernstian slope with −54.7 ± 2.90 mV/decade for K$^+$, −56.8 ± 5.87 mV/decade for Na$^+$ and −30.1 ± 1.90 mV/decade for Ca$^{2+}$. The intrinsic sensitivity of bare graphene towards K$^+$, Na$^+$ and Ca$^{2+}$ is also characterized. Although some of the individual sensors show sensitivity towards change in ion concentration due to defects on graphene channels[31], the averaged data shows negligible sensitivity (Supplementary Fig. 10). This further demonstrates the effectiveness of averaging over many devices to eliminate uncontrolled sensitivity that could originated from defects and contaminations. The averaged sensitivity of the ISM-functionalized sensor array is stable and repeatable over multiple measurements (Supplementary Fig. 11a). The robustness of the ion sensor array is further showcased by the negligible drift in sensitivity over a 6-month period (Supplementary Fig. 11b). The average response time towards K$^+$, Na$^+$, Ca$^{2+}$ are 7.4 ± 1.3 s, 5.9 ± 3.3 s, 5.1 ± 1.1 s (Supplementary Fig. 12), which are either faster or comparable to previously reported ion sensors, including commercial ones[32–34]. The feasibility of the sensor system's operation in complex biofluid is demonstrated through testing in artificial urine (AU) and artificial eccrine perspiration (AEP), where the relative concentration for K$^+$, Na$^+$, and Ca$^{2+}$ ions are within the 1–100 mM range. The molecule contents for AU and AEP are listed in Supplementary Table 2 and the

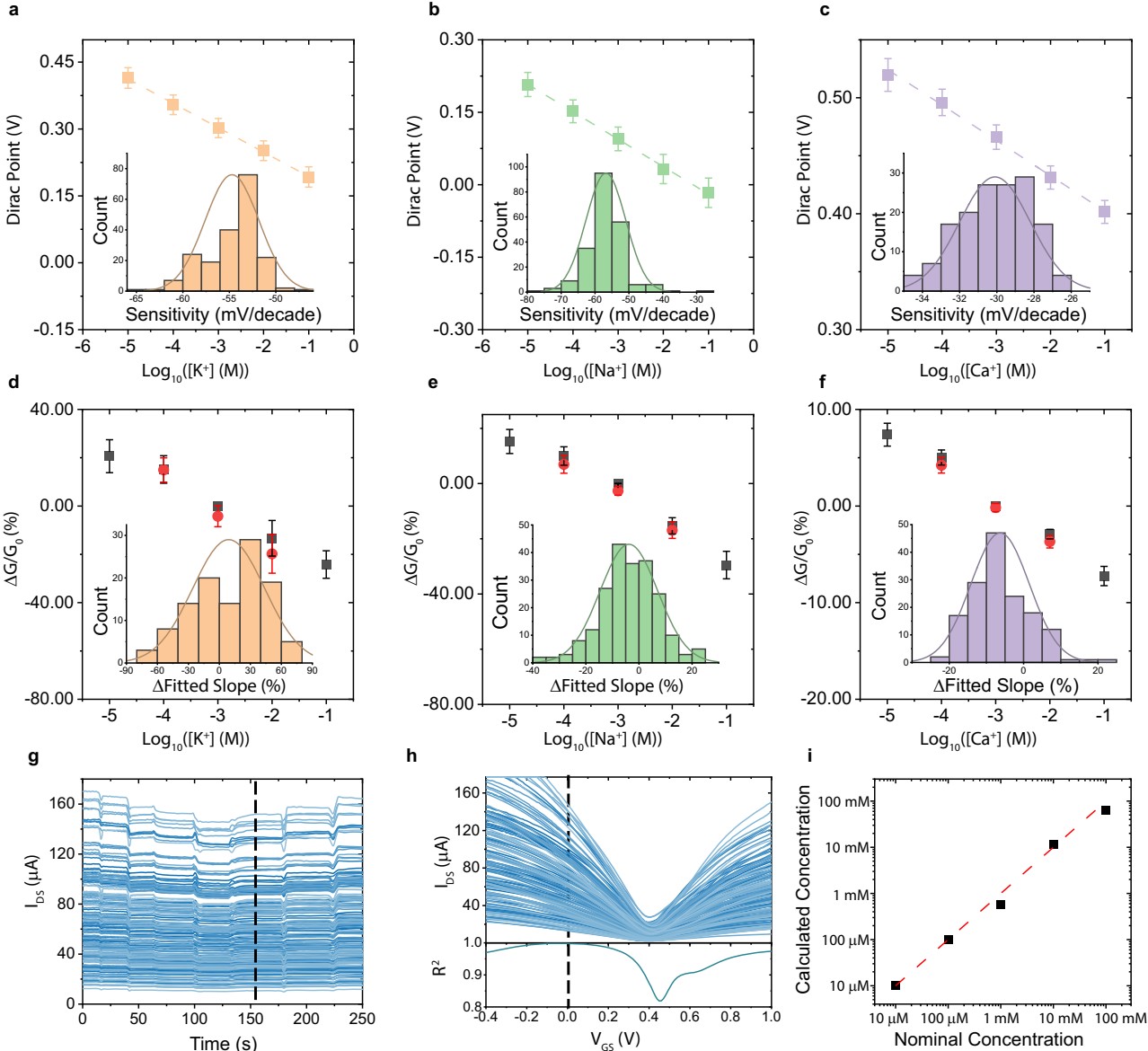

**Fig. 2 | ISMs-functionalized graphene sensing chip and profile matching calibration.** The minimum conduction point as a function of ion concentration for **a** $K^+$ ISM, **b** $Na^+$ ISM, and **c** $Ca^{2+}$ ISM functionalized graphene sensor chips. Error bars indicate the standard deviation of the Dirac Points of all the working devices on the chip. Dashed lines are the linear fits of the Dirac Points and the slopes of the fitted lines are the sensitivity. Insets depict the sensitivity distribution of individual devices on each chip. **d–f** The average change in channel conductance for $K^+$ ISM, $Na^+$ ISM, and $Ca^{2+}$ ISM functionalized sensing chip showing excellent reversibility over several magnitude change in $K^+$, $Na^+$, and $Ca^{2+}$ concentration. Black dots are measurements going from low ion concentration to high, while red dots are measurements going from high ion concentration to low. The percentage change in conductance is normalized

with respect to the data taken at 1 mM and the error bars are given by the standard deviation. Histograms in the insets show the reversibility distribution of individual devices on the sensor array. **g** Current transient response of all devices with $V_{GS} = 0$ V. The black dotted line indicates the source-drain current distribution at $Ca^{2+}$ concentration of 10 mM. **h** I–V characterization of all devices at $Ca^{2+}$ concentration of 10 mM and the correlation coefficients ($R^2$) obtained by matching the transient slices of 10 mM $Ca^{2+}$ to I–V characterizations at different $V_{GS}$ slices. **i** Calculated concentration at discrete intervals (black squares) using the profile-matching calibration method are plotted as x-values against the true concentration as the y-values. A fitted line (red dashed line) with a slope of 0.969 with an R-squared value of 0.996, indicating the effectiveness of the calibration method.

average sensitivities towards $K^+$, $Na^+$, $Ca^{2+}$ are plotted in Supplementary Fig. 13. The Nernstian sensitivity exhibits little change in the presence of other molecules and ions. This demonstrates the stability, selectivity, and compatibility of our sensors with real-world and complex samples for biomedical applications.

To investigate the reversibility of the sensing system, we measure the change in channel conductance versus ion concentration in Fig. 2d–f, where black dots are forward measurements (increasing ion concentration) and red ones are backward measurements (decreasing ion concentration). The sensor's reversibility is quantified by

calculating the percentage difference between slopes fitted to the forward measurements versus the backward measurements. Figure 2d–f insets show the reversibility of individual devices on the sensing chip. The average difference of fitted slopes is below 10% while the difference of an individual device could be over 80% in the worst case. Similar reversibility results were observed using the shift in Dirac Point instead of channel conductance (Supplementary Fig. 14). This again highlights that sensor metrics such as sensitivity and reversibility can be improved significantly by averaging over a large number of devices. Compared to other reported ion sensors, which normally have

less than five devices tested or reported, our ion sensing arrays exhibit excellent sensitivity, good reversibility, large detection range, and long-term stability (Supplementary Table 1). In comparison to the commercial ion-selective electrodes (ISEs), our sensor array system is able to detect ions with comparable sensitivity (Supplementary Table 1) but with enhanced portability. Graphene's high surface-to-volume ratio also makes it possible to have more than 200 devices within a small form-factor, something not possible when using bulky ISEs. The three-terminal transistor structure of the sensing units also adds gain and tunability compared to traditional ISEs. The biofluid compatibility and functionality in multi-ion detection, which will be discussed in latter section, make our sensor system a step closer to realizing reliable sensing for biomedical applications.

## Profile-matching calibration

Apart from using statistics to overcome the material and device variations to achieve stable, reliable, and highly accurate sensor behavior, the variations in the $I$–$V$ profiles between devices can be used to ease the system-level calibration. Here we propose a time-efficient calibration method based on the quasi-linearity in graphene's $I$–$V$ curves and one dataset gathered from the large sensor array representing the unique current level distribution among sensing units due to device-to-device variation. The Profile-Matching Calibration method is based on the fact that changes in concentration produce changes in the sensor operating currents $I_{DS}$ at a fixed gate voltage. As an example, first consider the current transient response and $I$–$V$ characteristic curves of a $Ca^{2+}$ ISM functionalized sensor chip. The black dotted line on the transient data (Fig. 2g) indicates a slice of operating current $I_{DS}$ at the 10 mM concentration at $V_{GS} = 0$, and the distribution of such current data due to inhomogeneities between the different devices in the array will be utilized and referred as "test current slice". The $I$–$V$ characteristic curves taken at 10 mM concentration can also be sliced at different $V_{GS}$ values, and the $I_{DS}$ distribution of each slice will be referred as "calibration current slice" (Fig. 2h, upper). We have a calibration current slice for each value of $V_{GS}$. The test current slice can then be compared to each calibration current slice using a standard optimization technique such as least squares fitting off-chip to find which slice of calibration current matches it the best (Supplementary Fig. 8). Since the test current slice was taken at $V_{GS} = 0$ at the same concentration as the calibration, the regression analysis will show a higher $R^2$ value with the calibration current slice closer to $V_{GS} = 0$ as shown in Fig. 2h.

Now, consider measuring with an unknown concentration. The corresponding test current slice taken at a fixed $V_{GS}$, can be mapped onto the calibration $I$–$V$ characteristic at the reference concentration in the same way as discussed above. The optimum calibration current slice is calculated by finding the value of $V_{GS}$ that provides the highest correlation coefficient $R^2$ between the calibration current slice and the test current. The relative voltage shift is the difference between gate voltage of the test current slice and that of the optimum calibration current slice. The solution concentration can then be readily calculated using the reference concentration, the voltage shift, and the sensitivity slope.

Calibration is essential for almost all sensing systems since sensors can deteriorate through wear, aging, and environmental influences[35]. Similarly, electrolyte-gated graphene field-effect transistors (EGFETs) will also need to be re-calibrated due to the unavoidable drift overtime because of graphene's property changes in ambient air (Supplementary Fig. 15). The methods typically used to calibrate EGFETs perform full $I$–$V$ characterization of the devices under multiple dilutions spanning the entire range of interest[36–38]. The profile-matching method however requires only a single measurement of $I$–$V$ characteristics using one calibration solution of known concentration. The calibration solution should be within the linear range of the device (10 μM–100 mM) in order to utilize the

quasi-linearity in graphene's $I$–$V$ curve. Since the sensitivity drift of the ion sensors is negligible over six months as shown in Supplementary Fig. 11, the sensors can be easily calibrated with the profile-matching approach to achieve the same sensing accuracy over multiple testing sessions.

The accuracy of this alternative measurement technique is investigated over several orders of magnitude change in $Ca^{2+}$ concentration as an example. Calculated concentrations are then compared to nominal concentrations capturing the combined accuracy of the graphene Ca ion sensors and measurement technique. Figure 2i shows that graphene Ca ion sensors quantify $Ca^{2+}$ concentration exceptionally well over several orders of magnitude with only one $I$–$V$ characteristics calibration as a reference.

## Highly integrated array for multiplexed sensing

We have demonstrated graphene-based sensor arrays with enhanced performance for $Ca^{2+}$, $Na^+$, and $K^+$ detection by averaging over the device-to-device variation. To further push the functionality of the sensing system, we demonstrate a sensing chip that integrates all three ISMs discussed earlier in the manuscript (Fig. 3a). The sensor chip is designed to have three separate regions. A Material Jetting 3D printing machine is used to deposit three ISMs to designated regions with spatial control of 100 μm and an average roughness under 5 nm (Supplementary Fig. 17). Details of the printing recipe can be found in Methods and the optical image of the sensor array after ISM printing is shown in Supplementary Fig. 1e. Figure 3b shows the $I$–$V$ characteristics of the chip measured in electrolytes with $K^+$, $Na^+$, and $Ca^{2+}$ ions after functionalization. In addition to the intrinsic variation in the $I$–$V$ characteristics of pristine graphene (Supplementary Fig. 3a), the $I$–$V$ curves from each type of ISM sensor also have their unique signature shapes, indicating that each ISM modifies the electrical properties of graphene in a different way[13,38,39]. Both $K^+$ and $Ca^{2+}$ ISMs heavily p-dope the graphene channel while $Na^+$ membrane shows a more neutral functionalization. The difference in the doping effect can be attributed to the molecular structure of different ionophores[40,41] (Supplementary Note 6). The color map of Dirac Points (Fig. 3a) shows a clear distinction between three ISMs.

Two sets of measurements were conducted using the integrated sensing chip. The first set, called "pure solutions," contains 15 electrolyte solutions with only one type of analyte ion with various concentrations spanning several orders of magnitude. The second set, called "mixture solutions," contains electrolyte solutions with multiple types of analyte ions in deionized water and will be discussed in the following section. Figure 3c shows the average shift in Dirac Points of the integrated chip tested in solution containing various $Na^+$ concentrations. Even though the sodium membrane exhibits the most significant change with respect to $Na^+$ concentration, there is also useful information embedded in the response of calcium and potassium ISMs. All three ISMs are highly selective and exhibit the highest sensitivity towards their target analytes. They also, however, exhibit some cross-sensitivity to other ions. Other researchers have observed similar behavior, but most have assumed absolute selectivity from ISMs and have ignored the cross-selectivity in their data analysis[38,42,43]. We perform principal component analysis (PCA) using Dirac Points of individual devices from measurements with the pure solutions as features to visualize the multivariate data under a lower dimensional space while preserving the largest variance. The first two principal components (PC) accounted for 92.1% of the total variance in the data. The score of PC1 and PC2 is plotted in Fig. 3d and the details of the PCA analysis are explained in Supplementary Note 7. The clusters of $K^+$, $Na^+$, and $Ca^{2+}$ ions are well-separated, indicating the sensor's ability to distinguish between different types of ions in electrolyte. Further separation can be achieved by using the ion sensitivities of individual sensors as the feature set as shown in Supplementary Fig. 20.

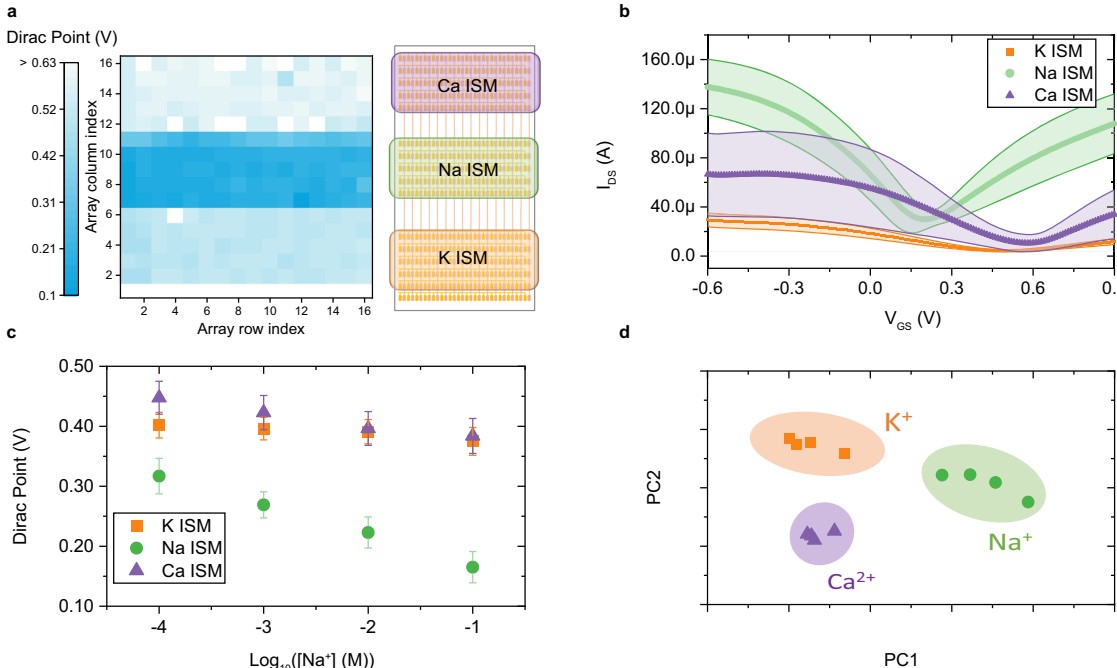

**Fig. 3 | Highly integrated sensing chip with all three ISMs. a** Schematic of the fully integrated ion sensing chip with all three ISMs (right) and corresponding color map showing the Dirac Point of each sensing unit. The average Dirac Point prior to functionalization is 0.15 ± 0.04 V. White pixels indicate nonfunctional cells due to the fabrication process **b** I−V characteristics of devices on an integrated sensing chip measured in an electrolyte solution containing 100 μM K⁺, 30 mM Na⁺ and 1 mM Ca²⁺ ions. The solid line is the average I−V characteristic and shaded region indicate an error band of plus or minus one standard deviation. Different colors indicate sensors functionalized with different ISMs. **c** Dirac Points of the integrated sensing chip in solution containing various Na⁺ ion concentrations. Error bars indicate plus or minus one standard deviation of the Dirac Points of all the working devices on the chip. **d** Principal component analysis (PCA) of the integrated sensing chip towards electrolytes comprised of different of ions using Dirac Points as feature vectors.

## Algorithm-enhanced sensing accuracy

Dirac Point is not the only parameter that has shifted upon the deposition of the membranes. The shape of the I−V curves also contains useful information about the functionalization and analytes that can be analyzed. Therefore, in addition to the Dirac Point, we also extract features from the I−V curves including maximum and minimum transconductance, which represent electron and hole currents, in an attempt to capture the relevant changes in graphene's physical properties. The full list of features and extraction methods are listed in Supplementary Fig. 21. Random Forest, a tree-based ensemble learning algorithm, was chosen to analyze the multidimensional features extracted from raw sensor signals because of its previous success in analyzing the biomedical sensor data[44]. This technique uses a robust ensemble learning approach to make the final decision through voting of predictions of many decision trees into a model but with less arbitrary hyperparameter search and tuning[45,46]. Such ensemble learning reduces variance and mitigates overfitting with internal cross-validation. Due to its tree-based nature, Random Forest is also much more interpretable and less data-hungry than complex learning frameworks such as deep neural networks, making it more suitable for the dataset from our sensing chip. Other classification algorithms that are commonly used for disease prediction problems[47], including latent Dirichlet allocation (LAD), support vector machine (SVM), K-nearest neighbors (k-NN), and Gaussian process (GP), are also compatible with the dataset gathered by the multiplexed sensing chip as shown in Supplementary Fig. 23. The model implemented by the Random Forest algorithm has comparable or better performance relative to other classification algorithms.

We first trained a classifier with data from the pure solution tests and we are able to achieve 97.6% accuracy in classifying the presence of ion types as shown in Fig. 4a. The feature importance of the trained model is analyzed using the game theory-based interpretability

algorithm, SHAP (Shapley Additive exPlanations)[48,49] and the top four most important features are shown in Fig. 4b. The Dirac Point of calcium ISM and sodium ISM contribute the most in the ion type classifier while potassium ISM provides less information. This gives insights in designing a large area cross-reactive array for specific tasks. For simple tasks such as classification between K⁺, Na⁺, and Ca²⁺ ions, only two types of functionalization would suffice.

The ion concentration can also be predicted. Since the ion concentration of testing solutions varies across several orders of magnitude, we use a classification approach instead of a traditional regression model. The resulting models produce classification accuracies of 83.1, 86.9, and 77.9% for K⁺, Na⁺, and Ca²⁺ ion concentrations, respectively (Supplementary Fig. 22). The models can also indicate if the incorrect model was used for an unknown solution. For example, if the Ca model is used for sodium solution, the model will output concentration class "zero".

Although it is important to determine the ion type and concentration in pure solutions, real-world applications require high performance in solution with multi-ions. Therefore, a second set of experiments was performed using multi-ion solutions, where all three types of ions are present simultaneously. Multiclass Random Forest classifiers are trained to predict the concentration of K⁺, Na⁺, and Ca²⁺ and the accuracy for each ion is 90.6, 82.6, and 61.7%, (Fig. 4c−e). From the machine learning perspective, the Ca²⁺ concentration prediction task is harder since the representation learned from the selected features may have less explanation power to high Ca²⁺ concentration values (note: 10 and 100 mM) even though the model can do relatively well at lower concentrations. These results show the potential for these multiplexed sensing chips to perform well in complex, real-world environments.

In addition to complex electrolyte profiles, biological fluids also contain antibodies, antigens, and hormones that introduce more

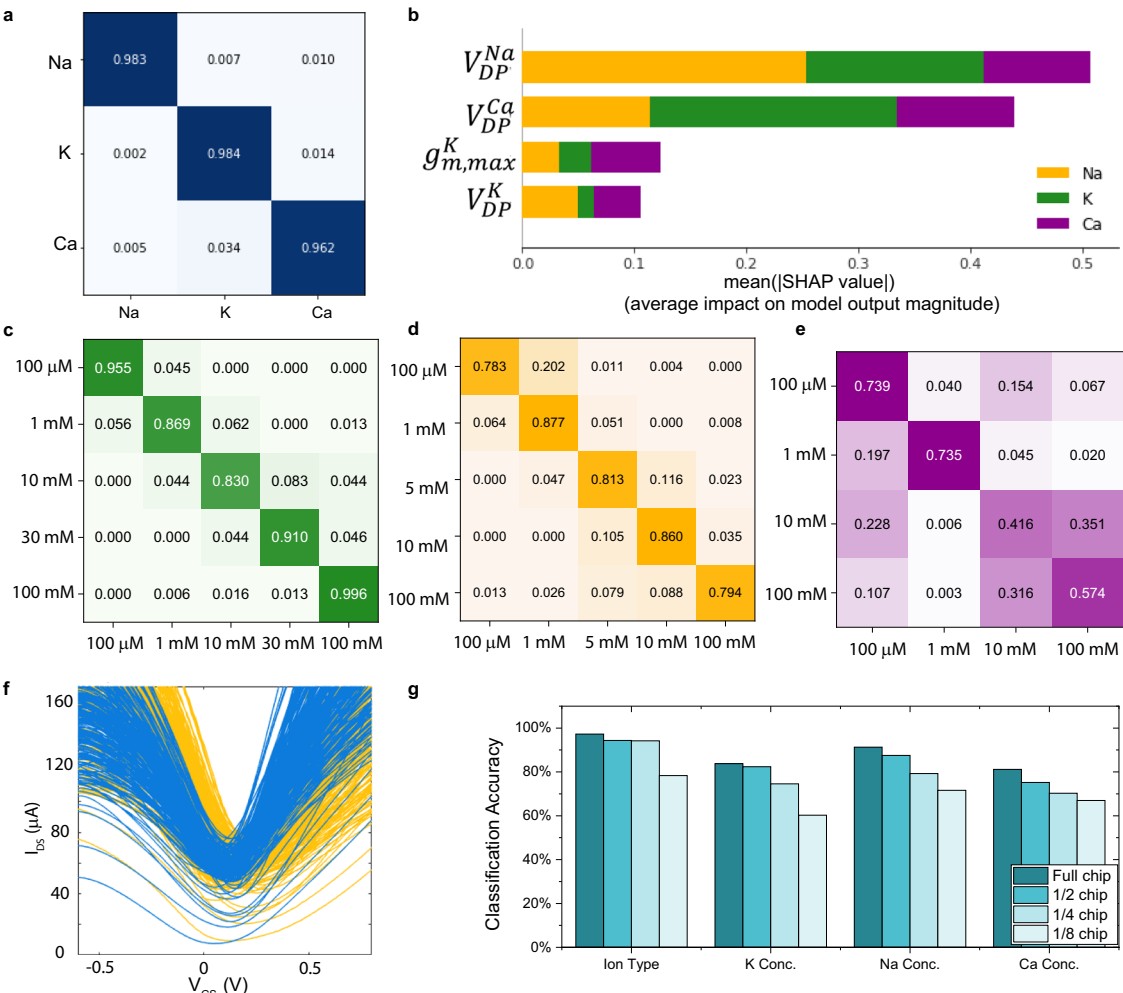

**Fig. 4 | Random Forest Model using data from highly integrated sensing chip.**
**a** Confusion matrix of the ion-type classifier. **b** Top four SHAP (Shapley Additive exPlanations) values for the ion-type classifier. $V_{DP}^{Na}$, $V_{DP}^{Ca}$ and $V_{DP}^{K}$ represent the Dirac Points of devices with $Na^+$ ISM, $Ca^{2+}$ ISM, and $K^+$ ISM functionalization respectively. $g_{m,max}^{K}$ represents the maximum slope of devices with $K^+$ ISM functionalization. **c**–**e** Confusion matrix of the multi-output classification model for $K^+$, $Na^+$, and $Ca^{2+}$ ion concentration in multi-ion solutions. **f** I–V characteristics of graphene sensor array tested in water (yellow curves) and in human serum (blue curves). **g** Impact of chip usage on classification accuracies for the pure solutions test.

confounders into the analysis. The graphene-based sensors will behave differently in terms of current level, Dirac Point, cross-sensitivity, detection limits and so on (Fig. 4f). The multiplexed sensor array analyzes the collective response of an array instead of assuming absolute sensitivity or selectivity of one device. Models can be readily calibrated or re-trained using real-world biological samples in future studies. Here we simulate how the system would be used in a real-world setting, where biofluids are analyzed using the concentration profile of the $K^+$, $Na^+$ $Ca^{2+}$ ions as biomarkers for electrolyte imbalance disorders. We re-categorized the mixture solutions and trained the model to predict the mixture concentration profile instead of the concentration of individual ions (Supplementary Table 4). The confusion matrix of the result model is shown in Supplementary Fig. 24. We achieve an average accuracy of 84.7% for the classification of ion concentration profiles with solutions containing multiple ion types. The model is less confident at identifying higher calcium and higher potassium. This could be due to the imbalanced training data in these classes, i.e., there are fewer observations with a higher calcium/potassium level that make the learning more difficult. The lower confidence on concentrated $Ca^{2+}$ solutions is also observed in the $Ca^{2+}$ ion concentration model shown in Fig. 4e. A possible reason could be the intrinsic lower Nernstian slope for bivalent ions and the choice of features. Quantifying higher calcium concentrations could be further

improved by carefully redesigning the functionalization matrix and optimizing feature selection. This analysis demonstrates the possibility of coupling the multiplexed sensing array with machine learning models to achieve fast and high accuracy electrolyte imbalance-related disease diagnoses. The performance of the model could be greatly improved with a larger, balanced training dataset. The model could also be readily re-trained towards specific diseases using training data collected with real biofluid from patients in a clinical trial.

We further analyze the importance of sensor redundancy and sensor multimodality by evaluating the model accuracy when sampling from a smaller device pool, which mimics more conventional approaches of 2D material-based sensors using fewer devices. Figure 4g shows the accuracy of different tasks sampling from 1, 1/2, 1/4, and 1/8 of the total devices using the pure solution set. Compared to the accuracies stemming from 25 devices (1/8 chip), which already contains more devices than most works on ion sensors (Supplementary Table 1), we show that large scale integration, with over 200 devices, can improve the accuracies more than 20 percentage points. For complicated tasks, it is necessary to have a large set of sensors available for training in order to achieve the desired accuracy. It is possible to occasionally have comparable or slightly better model performance due to the random dropping of the training samples while decreasing the number of sensors. It is also possible that data

distributions between the test samples and the training samples after dropping some sensors are similar compared with the whole training set, and therefore will not produce improvement even after adding more samples in the training. Therefore, large device-to-device variations, which are typically detrimental in most advanced-material-based sensors, can actually be beneficial in our approach. The benefits of sensor redundancy are also quantified by analyzing the relationship between device number and accuracy when performing the profile-matching calibration methods. 50,000 synthetic test current responses and $I$–$V$ characteristics are generated according to empirical cumulative distribution functions (ECDFs) derived from experimental data so as to mimic experimental data as closely as possible. The result is depicted in Supplementary Fig. 25 and shows tightening of 95% confidence intervals from ±50% to within ±10% of the $Ca^{2+}$ concentration. The tightening of the confidence intervals further demonstrates the importance and effectiveness of high-density sensor arrays on enhancing sensor performance.

We also investigate the role of each functionalization and further improve the model's interpretability by performing ablation tests. Supplementary Fig. 22d shows the testing error if one of the ISMs is removed. For ion type classification, removing the calcium or potassium membrane does not increase the error significantly but removing the sodium ISM yields a larger error. This aligns with the SHAP interpretation in Fig. 4b where the Dirac point of the $Na^+$ ISM is the most significant factor for the ion classification. For ion concentration prediction tasks, we find that removing the corresponding ISM leads to the largest performance drop except for calcium. The result also confirms that models for potassium and sodium concentration prediction have better performance than calcium, since predicting the $Ca^{2+}$ concentration requires the information from other ISMs, or even more information is needed for constructing a better data representation for modeling. Similar results were found with the mixture solution test as shown in Supplementary Fig. 22e.

## Discussion

In this work, we develop a highly integrated graphene-based sensing platform that addresses and overcomes the current limitations in 2D materials technology and achieves high performance and enhanced functionality. The scalable fabrication process provides a promising way for future sensors to achieve both excellent performance and low cost. The portable sensing platform uses more than 200 graphene sensors to demonstrate accurate detection of calcium, sodium, and potassium ions in simple electrolytes, artificial urine, and artificial eccrine perspiration. The proposed sensing platform combines a statistically significant sample size and a calibration method to overcome device-to-device variation and demonstrates excellent sensitivity and reversibility. The Random Forest algorithm was used to demonstrate accurate ion type classification, concentration prediction, and potential applications in electrolyte imbalance-related disease diagnostics. The importance and effectiveness of combining a large dataset with statistics and machine learning was demonstrated in terms of sensor performance enhancement. Our sensing platform can be readily adopted for other analytes of interest as well as with other advanced 2D materials to realize accurate and reliable multiplexed sensing for biomedical applications.

## Methods
### Device fabrication
The transducer part of the sensor platform described in this paper is an array of functionalized graphene-based EGFETs fabricated on a disposable glass slide. Fabrication of the array begins with piranha cleaning a 200 μm thick 4-inch Corning willow glass (MTI Corporation). The substrate was coated with 25 nm aluminum oxide using atomic layer deposition to enhance adhesion in subsequent photolithography steps. A layer of Ti/Au (5 nm/150 nm) was deposited using

electron beam deposition to form the row contacts of the sensor array. A 30 nm layer of aluminum oxide was then deposited as interlayer dielectric using atomic layer deposition. Openings were etched into the interlayer dielectric using a $BCl_3$ plasma to allow contact between the first and second metal layers in the array where appropriate. A second metal layer of Ti/Au (5 nm/150 nm) was then deposited using electron beam deposition. Graphene coated with poly(methyl methacrylate) (PMMA) (ACS Materials Trivial Transfer Graphene™ 1 cm × 1 cm) was transferred on the substrate to cover the entirety of the array. The graphene film was single-layered with minimal defects as shown in the Raman spectrum in Supplementary Fig. 2a, b. The chip was baked at 60 °C for 30 min and 130 °C for 2 h. This allows PMMA reflow and enhances adhesion between the graphene and substrate. The sensor array chip was then immersed in acetone for several hours to remove the PMMA. The chip was subsequently annealed at 350 °C in 400 sccm Ar and 7000 sccm $H_2$ to reduce PMMA residue and further enhance adhesion between the graphene and substrate. Graphene sensors were isolated from each other using oxygen plasma and a patterned PMGI/SPR700 resist stack as a mask. Photolithography is used to reduce cost while achieving higher throughputs. The resist stack was then removed by immersion in $N$-Methyl-2-pyrrolidone (NMP) for several hours. The chip was spin-coated with SU-8 and openings were defined over the graphene channel regions and contact leads. The whole process can be easily scaled up for 6 or 8-inch process and the cost will be further reduced. The Raman Spectrum after fabrication is shown in Supplementary Fig. 2c and the $D/G$ band mapping of a 20 μm × 20 μm graphene channel is shown in Supplementary Fig. 2d. Insert in Supplementary Fig. 2c shows the distribution of $I_{D\_band}/I_{G\_band}$ ratio sampled from 20 devices across the array. The low D band intensity indicates low density defects and minimal damage to the graphene channel.

### Measurement setup
All electrical measurements were performed using the custom-built measurement system (Fig. 1b, c) at room temperature. The measurement system makes use of an Atmel SAM3X microcontroller with an 84 MHz clock, which enables very high-speed data acquisition. Dual 12-bit digital-to-analog converters (DACs) are employed to vary the $V_{DS}$ and $V_{GS}$ voltages applied throughout the sensor array appropriately depending on the measurement configuration (e.g. $I$–$V$ sweep, transient $I_{DS}$). The microcontroller is paired with a custom printed circuit board (PCB) designed to precisely match the input and output ports of the microcontroller in order to achieve an overall compact form factor and portable measurement system (about the size of a cell phone). The custom PCB includes 16 transimpedance amplifiers (one for each column of the sensor array) that make use of 2-stages along with low-noise resistors and operational amplifiers to achieve an overall gain of 10,000 $V/I$ so that μA (and sub-μA) sensor currents can be amplified to the appropriate voltage range and measured with very high accuracy using a 12-bit analog-to-digital converter (ADC).

Row and column selection is performed using 16×1 bidirectional analog multiplexers with low on-resistance (2.5 Ω) to minimize distortion of the applied $V_{DS}$ voltages and sensor current readouts. One analog multiplexer is used to apply $V_{DS}$ along a single row of devices. The resulting column currents are all continuously amplified using the 16 transimpedance amplifiers. The second analog multiplexer is used to rapidly switch which transimpedance amplifier output is applied to the ADC for readout. After all column currents have been read out, $V_{DS}$ can be applied to the subsequent row and the process is repeated. In this way, we are able to rapidly scan the entire array of devices. Low dropout regulators are employed so that the entire measurement system can be conveniently powered using a single universal serial bus (USB) power supply. All measurement instructions and results are transmitted via USB as well.

For $I$–$V$ sweep measurements, the drain-source voltage $V_{DS}$ was held constant and the gate-source voltage $V_{GS}$ was swept from −0.6 to 0.9 V in 10 mV increments. A 10 s hold time was used before the gate-source voltage $V_{GS}$ was swept at a rate of 10 mV/500 ms. This provides adequate time for charged species to migrate and reach steady-state before measurement. All measurements were conducted under ambient conditions at room temperature. Solutions were prepared by serial dilution of 1 m aqueous $K^+$, $Na^+$, and $Ca^{2+}$ solution over several orders of magnitude to provide a variety of concentrations: 100 mM, 10 mM, 1 mM, 100 μM, 10 μM. The transient response was investigated by dipping the graphene sensor array in each dilution for approximately 20–30 s. The experiment starts with the lowest ion concentration to reduce the potential for altering solution concentrations due to cross-contamination. For the integrated chip, additional mixture solutions were prepared. The full list of solutions tested in this paper are listed in the Supplementary Information Tables 3, 4.

## Functionalization preparation and deposition

Ion-selective membrane solutions were made by mixing of 2-nitrophenyl octyl ether (NPOE), high molecular weight polyvinyl chloride (PVC), the specific ionophore, and potassium tetrakis(4-chlorophenyl) borate (KTCB). Calcium ionophore II (ETH 129), sodium ionophore X, and valinomycin were used for calcium, sodium, and potassium ISMs, respectively. The mixtures were dissolved in 6 mL of tetrahydrofuran (THF), which is approximately 85% by weight. All materials were bought from Sigma-Aldrich. The recipes for each ISM are listed in detail in Supplementary Information Table 5. For individual ion sensing arrays, the ISM solution was spin coated over the entire array at 1500 rpm for 120 s for and allowed to air dry. The deposition of the multiple functional layers on a single integrated sensing chip was performed using a jet valve droplet dispensing system (Pico Pulse, Nordson EFD). By modulating applied pressure, stroke, nozzle close and open times, actuation frequency, and nozzle size, we were able to fine-tune the size of the dispensed liquid droplets, and hence the dimensions of each functional layer. We adjusted the applied pressure to 20–30 psi and the stroke to 60–70% (percentage of voltage drop during the valve actuation) with the closed valve voltage of 100 V. The membrane uniformity over an area of 2.5 mm × 0.5 mm is characterized and shown in Supplementary Fig. 18. Three different functional layers were dispensed in ~1000 μm lines respectively; droplets were dispensed 0.2 mm apart at the frequency of 500 Hz from a 50 μm nozzle (7362574, Nordson EFD).

## Biofluids

Two types of artificial biofluids used in the study are Artificial Eccrine Perspiration (BZ112) and Artificial Urine (BZ186) purchased from Bio Chemazone INC. The human serum used in the study is sterile-filtered human male AB plasma (H4522) purchased from Sigma-Aldrich Co. LLC.

## Synthetic responses

ECDFs were used to generate transient responses and $I$–$V$ characteristics. Extending population size allows for more thorough statistical analysis. It also remedies statistical issues arising when the randomly generated sample size approaches the population size. Further details regarding the data synthesis process can be found in the Supplementary Notes 4, 5 and Supplementary Fig. 16. For each sample size, the corresponding number of transient responses and $I$–$V$ characteristics are randomly sampled from the overall population size. This captures the randomness associated with fabricating individual sensor arrays with N graphene Ca sensors. $Ca^{2+}$ concentrations are then calculated to capture the randomness in measurement accuracy for a sensor array with N graphene $Ca^{2+}$ sensors. This process is repeated at each sample size 1000 times in order to generate a distribution and capture the randomness associated with measurement accuracy as a function of sample size. These distributions allow the mean and 95% confidence intervals to be calculated for $Ca^{2+}$ concentrations as a function of sample size. This quantifies the benefits of having redundancy in sensors.

## Data processing and modeling

We categorized the sensors with respect to their functionalization into three groups and randomly sample a device from each group as a set of training data. Selected features (Supplementary Fig. 21) were extracted and an additional 5% Gaussian noise was added to feature vectors to prevent overfitting. Then all input features were normalized before feeding into the algorithm. The tree-based ensemble learning algorithm, Random Forest, was utilized to conduct the ion and concentration classification tasks due to its capacity to identify non-linear relationships between selected features. We set the number of trees in the forest to 100, the maximum depth of the tree to 20, and used the Gini impurity to measure the quality of tree splits. For each task, we did five independent experiments with the random 80/20 train/test split, and computed the mean, standard deviation, and 95% confidence interval of the classification accuracy. All input features were normalized before feeding into the algorithm. For the feature contribution analysis, we adopted Tree SHAP (TreeExplainer)[48,49] to compute the SHAP values for interpreting the output of Random Forest classifiers. We used the scikit-learn package to implement the data preprocessing, Random Forest model training, and evaluation, and used the SHAP package to investigate feature contributions. All experiments were conducted in Python version 3.6.

## Data availability

All data supporting this study and its findings are available within the article, its Supplementary Information, and associated files. Any source data deemed relevant is available from Figshare under accession code 10.6084/m9.figshare.20399175 and from the corresponding authors upon request.

## Code availability

All code that supports the findings of this study is available from the corresponding authors upon request.

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

## Acknowledgements

This project was financed by the Center for Integrated Quantum Materials under NSF grant DMR-1231319 (to M.X., C.M., and J.Z.) and by the U.S. Army Research Office through the Institute for Soldier Nanotechnologies, under contract number W911NF-13-D-0001 (to M.X., C.M., M.H., and E.M.). M.X. and Y.L. acknowledge MathWorks Engineering Fellowship. We would like to thank Dr. Yuxuan Cosmi Lin for proofreading and commenting on the manuscript.

## Author contributions

T.P. and C.M. conceived the project. M.X. and C.M. designed the study and prepared the manuscript. X.M., Y.L., J.Z., C.M., S.-X.L.L., M.H., and E.M. fabricated the devices. M.X. and C.M. performed the electrical measurements. M.X prepared the test solutions and functionalizations. C.M. conducted profile-matching calibration. W.W., M.X., and A.-Y.L. conducted machine learning. T.P. and J.K. directed all experimental research and supervised this work. All authors analyzed the results and implications and commented on the manuscript at all stages.

## Competing interests

The authors declare no competing interests.

## Additional information

Mantian Xue or Tomás Palacios.

**Peer review information** *Nature Communications* thanks Wangyang Fu
and the other, anonymous, reviewer(s) for their contribution to the peer
review of this work. Peer reviewer reports are available.

