## [Peer Review File · Nature Communications]

Integrated biosensor platform based on graphene transistor arrays for real-time high-accuracy ion sensingREVIEWER COMMENTS

Reviewer #1 (Remarks to the Author):

- What are the noteworthy results?

- In this manuscript, a highly integrated sensor based on graphene transistor arrays is constructed, demonstrating good capacity in the multi-channel measurement of different analytes on the same sensor.
- With the profile-matching calibration, the sensors can be easily calibrated, and the concentration of ions in an unknown sample could be quantified with only one reference solution, reducing the workload in the ion sensing effectively.
- By using PCA, K⁺, Na⁺ and Ca²⁺ ions can be well distinguished by the sensor, circumventing the cross-reactivity of each ISMs. With the random forest algorithm, different metal ions are well classified and different ion concentrations are also relatively well predicted.
- The portable sensing platform also demonstrated its ability to be operated in complex biofluids, such as artificial urine and artificial eccrine perspiration.
- The test protocol is well optimized. Different factors which may affect the test accuracy such as temperature, hold time, and test sequence, were investigated carefully to eliminate the signal drift in the consecutive tests.

- Will the work be of significance to the field and related fields? How does it compare to the established literature? If the work is not original, please provide relevant references.

- Yes. The work provides a solution to improve the test accuracy in biosensing with the 2D-material based devices and a new way to overcome the device-to-device variation. The highly integrated sensor with more than 200 devices guarantees the abundance of units in the test and reduces the requirements of device uniformity. The application of statistical analysis and machine learning algorithm further improve the test accuracy. The portable sensing platform also provides one step closer to achieving its real-life applications.

- Does the work support the conclusions and claims, or is additional evidence needed?

- What is the physiological significance of studying K⁺, Na⁺ and Ca²⁺? What are the relative concentrations of these ions in AU and AEP?
- The abundance of sensors in the test is important for the improvement of test accuracy in this work. It's really a perfect work to build such a highly integrated sensing platform, but workload and cost should also be balanced in a project. The authors should also discuss these two aspects in the paper. In addition, will the sensitivity/selectivity of the same sensor decrease after multiple tests? What about the shelf-life (although the sensitivity kept well in 10 h in Figure S7b)? How robust are the ISMs?
- Since there are more than 200 devices on the same sensor, the multi-channel measurement is important for the data collection. How many devices could be measured simultaneously in the testing process? How long does it take to finish the measurements on all the 215 devices in a test?
- According to the equation 3, the sensitivity is limited by Nernst limit (around 59 mV/decade). The sensitivity of some devices was actually higher than this value (shown in Figure 2a-c, Figure 3a, and Figure S7a). The authors should explain the deviations of sensitivity values, although some sensors with even higher sensitivity have been reported.
- In Algorithm-enhanced sensing accuracy section, the authors should clarify if the multi-ion solutions used in the second set of experiments were in water or AU/AEP. It would be better if the authors could provide the sensing performance and classification results with real-world biological samples to support the practical significance of the sensing platform. Or the authors should at least discuss on how the sensing performance and data analysis could be affected due to the complexity of real-world biological samples.

- Are there any flaws in the data analysis, interpretation and conclusions? - Do these

prohibit publication or require revision?

- For the PCA, how many features were applied for the classification? The authors should clarify these features in the manuscript.
- For the artificial urine (AU) and artificial eccrine perspiration (AEP), the authors may make their recipes opposite in Table S2 since there should be urea and uric acid in AU.
- As shown in Figure 2d-f, the sensitivity changed with the opposite test sequence (from high concentration to low concentration). Could it be explained with the cross contamination? At the same time, there are only three data points (red one) in Figure 2d-f. What about the relative conductance changes at other two concentrations? Lastly, only $\Delta G/G$ was applied for the investigation of reversibility in the manuscript, the Dirac point change from high concentration to low concentration should also be plotted.

- Recommendation is publication after minor revisions.

- Is the methodology sound? Does the work meet the expected standards in your field?

- Yes.

- Is there enough detail provided in the methods for the work to be reproduced?

- The authors claimed that only one reference solution was need for the determination of ions in a unknow sample with the profile-matching calibration. Is a reference solution always essential for the calibration in each test, or not needed any more after the calibration of sensor with the reference? In addition, what was the concentration of reference for the calibration in Figure 2i? Should the concentration of reference always fall in the linear range?
- In Figure 4f, what was the concentration of each ion for the test? Was the mixture solution used for the test?
- For the classification of different ion concentrations in the mixture solution, the confusion matrix is still limited to each ion, which contains five classes. Could the authors provide the confusion matrix with fifteen classes? (Three types of ions and five different concentrations).
- The errors in units should be corrected. "mm" should be "mM" in page 15, 18 and 30.

Reviewer #2 (Remarks to the Author):

Xue et al. reported a field-effect sensing platform based on more than 200 integrated graphene ion sensors. The authors claimed that they overcome the limitation of relatively large device-to-device variation in 2D materials, and achieved high-performance ion sensing by adopting new calibration methods as well as Random Forest algorithm. The manuscript is well-written and the technique seems to be reasonable. Given the research tackles one of the most important limitations/problems in biochemical sensors based on non-uniform 2D materials, in my opinion this paper should be accepted with minor revision.

1. In Supplementary Fig. 1f, the performance of the device is quite different. I suggest giving statistical data on resistance or carrier mobility for the same batch of devices for comparison. There are only few characterizations on the surface morphology and quality of the graphene devices. The authors should show optical images of individual devices (as-fabricated and functionalized).

2. Material jet printers are used to deposit various chemicals onto the sensing area with precise lateral control. To what extend the printers can control the uniformity of the functionalization?

3. Fig. 3c gives the ion-selective results for ion-selective membranes with different membrane functionalizations, where Ca ISM exhibits a non-selective sensing response towards Na⁺ ions. Did the authors test the ionic sensing response of the as-fabricated graphene devices? It is worth noting that previous works (Nano Lett. 2011, 11, 3597

and Adv. Mater. 2017, 29, 1603610) demonstrated that intrinsic defects, as well as possible contaminations introduced during device fabrication and storage, might lead to uncontrolled sensitivity.

4. In my opinion, a reliable and stable operation of the graphene devices is the key. The authors should provide long-term stability measurements of as-fabricated and functionalized graphene sensors.

Reviewer #3 (Remarks to the Author):

In this manuscript, the authors present a novel approach to address current challenges in 2D material-based sensing devices such as material quality variability, device uniformity, high performance and enhanced functionality in ion classification application.

The proposed approach reduces the requirements on material quality and device uniformity. To this end, the authors fabricated sensor arrays consisting of 16×16 graphene devices to provide more than 200 working sensing units for each chip, and configured them to achieve multi-ion sensors by modifying their surface with three different ion-selective membranes (ISMs). Moreover, machine learning algorithms such as the Random Forest algorithm, were applied to demonstrate the enhanced functionality of accurate ion type classification and concentration prediction. The developed platform demonstrates near ideal sensitivity, excellent reversibility, and a large detection range for each type of sensor despite the large non-uniformity of the individual devices. The sensing platform could be readily adopted for other analytes of interest as well as with other advanced 2D materials that also suffer from the same issues (e.g., device-to-device variations) to realize accurate and reliable multiplexed sensing in biomedical applications.

The manuscript is interesting and generally fulfils the scope of Nature Communications. However, in its present form it can not be recommended for publication. To provide an adequate contribution to the journal, the authors have to take into account the following points in a major revision:

1. On page 3, the full name of the abbreviations "PEDOT" and "PSS" should be given at their first appearance in the manuscript.
2. On pages 3 and 4, the two paragraphs, "Here we demonstrate a novel" and "In this paper, we fabricate arrays of" are not well organized thus hard to understand. There are a lot of redundant elements, such as the sentence "Instead of focusing on the improvement of intrinsic material quality, fabrication uniformity and surface functionalization, we demonstrate ways to overcome the large degree of variability of advanced materials by developing a high-density graphene-based sensor array, thus significantly reducing the requirements on material quality and device uniformity. ". The section following "thus" is repeated as the section following "Instead of", etc.
3. The authors present the Raman characterization before and after the sensor fabrication. Did the authors also perform the Raman mapping characterization for the graphene film after sensor fabrication procedure? This could be helpful to show the quality of whole graphene film after fabrication.
4. In the manuscript, the authors point out that there are $16 \times 16 = 256$ sensors in the array. However, only 215 working devices were functionalized and presented, as shown on page 7. The authors should explain why not the 256 sensors are utilized and ways to overcome this limitation.
5. In the section "Highly integrated Array for Multiplexed Sensing": The authors should comments on the observation "Na membrane shows a more neutral functionalization". Which are the underlying mechanisms?
6. In Figure 3(d), the author show the PCA score plot of the chip sensitivity, but the introduction of the input features (multivariate) are not explained in the manuscript. This should be improved.
7. In the "Algorithm-enhanced sensing accuracy" section, the authors present the

classification accuracy implemented by the random forest algorithm. Did the authors perform the classification with other classifier algorithms, such as LDA, SVM, KNN, etc? Which results can be expected?

8. On page 15, the authors conclude the accuracies of 1/8 chip is larger than most of the reported sensors. The authors should give more detailed results of the other works and include the necessary citations.

9. The author performed all the measurement on artificial bio-fluids. Did the authors perform the measurement on real bio-fluids, and which are the main possible challenges related to this? Is the applied technology and its underlying methods sufficiently robust?

10. From the manuscript it seems that the authors conducted all measurements on a single chip. There is no suggestion in the manuscript that more than one chip was reproduced. How is the reproducibility?

11. The author show several I_{sd} - V_{gs} characteristics. Have the author conducted any characterization of the I_{sd} - V_{sd} for all the sensors? How large is the resistance range of all the sensors after fabrication (before applying liquid-phase environment)?

The following minor typographic corrections need to be corrected:

1. On page 5, in the sentence "The quality of the intrinsic graphene film is analyzed by Raman Spectroscopy as shown in Supplementary Fig. 1b-c" , obviously, Supplementary Fig. 1b is not the Raman characterization result, it should be "The quality of the intrinsic graphene film is analyzed by Raman Spectroscopy as shown in Supplementary Fig. 1c-d".

2. Page 13 : "The score of PC1 and PC2 is plotted in Fig. 3e. ". It should be "Fig. 3d".

3. Page 17 - typo, "The graphene film was single-layered with minimal defects as shown in the Raman spectrum in Supplementary Fig.1b-c", it should be "Fig.1c-d"

4. Caption of Supplementary Figure 1 (c), "c Raman spectrum the intrinsic graphene sheet before any fabrication, here "of" is missing between "spectrum" and "the".

Reviewer #4 (Remarks to the Author):

It is a very well written paper but real time monitoring of real sample can't be demonstrated by using simply urine samples... Two questions: sensing in plasma fluids are in presence of a high (9g/L) of NaCl concentration and in presence of complex mixture of proteins. The high concentration of salt should decrease the sensitivity of FET sensing scheme, how do the authors aim to overcome this? Same question in a complex fluid with grams per millilitre of serum albumin, sugar, etc.

Reply to reviewer 1's remarks:

Overall Remarks:

- In this manuscript, a highly integrated sensor based on graphene transistor arrays is constructed, demonstrating good capacity in the multi-channel measurement of different analytes on the same sensor.
- With the profile-matching calibration, the sensors can be easily calibrated, and the concentration of ions in an unknown sample could be quantified with only one reference solution, reducing the workload in the ion sensing effectively.
- By using PCA, K⁺, Na⁺ and Ca²⁺ ions can be well distinguished by the sensor, circumventing the cross-reactivity of each ISMs. With the random forest algorithm, different metal ions are well classified and different ion concentrations are also relatively well predicted.
- The portable sensing platform also demonstrated its ability to be operated in complex biofluids, such as artificial urine and artificial eccrine perspiration.
- The test protocol is well optimized. Different factors which may affect the test accuracy such as temperature, hold time, and test sequence, were investigated carefully to eliminate the signal drift in the consecutive tests.
- The work provides a solution to improve the test accuracy in biosensing with the 2D-material based devices and a new way to overcome the device-to-device variation. The highly integrated sensor with more than 200 devices guarantees the abundance of units in the test and reduces the requirements of device uniformity. The application of statistical analysis and machine learning algorithm further improve the test accuracy. The portable sensing platform also provides one step closer to achieving its real-life applications.
- Recommendation is publication after minor revisions.

Our response:

We thank the reviewer for the acknowledgment on the novelty and impact of our work. We also thank the reviewer for the constructive comments, which have significantly helped to increase the clarity of our manuscript and highlight its contributions to material science.

1. What is the physiological significance of studying K⁺, Na⁺ and Ca²⁺? What are the relative concentrations of these ions in AU and AEP?

Our response:

We appreciate the reviewer's comment. Electrolyte imbalance in biofluid such as sweat, blood, urinate can be indicators for individual's physiological state. Potassium is one of the most important cations in body electrolyte. It helps regulate fluid balance, muscle contractions and nerve signals. Potassium disorder can be caused by diuretic use, gastrointestinal losses, kidney disease and hyperglycemia^{R1}. Sodium ion concentration is another crucial component for potential disease diagnosis. Disorder in sodium ion concentration is one of the most common electrolyte disturbances. Server alternation of ionized sodium level in biofluid is associated with considerable morbidity and mortality^{R2}. Sodium ion disorder are often iatrogenic therefore fast and easy detection or monitoring of sodium concentration can greatly help diseases diagnosis efficiency. Calcium is also an essential component for human's minerals homeostasis^{R3}. Ionized calcium plays

an important role in mediating vascular contraction and vasodilation, muscle contraction, nerve transmission, and glandular secretion. Imbalanced calcium ion concentration could be indicators for parathyroid condition, kidney disease, thyroid disease and so on.

A detailed discussion on the physiological significance of K^+ , Na^+ and Ca^{2+} ions and relevant references are mentioned in the main text as: *“They help regulate fluid balance, muscle contractions, nerve system transmissions, and glandular secretion. A better understanding and instant monitoring over these ions are essential for evaluation of patient’s physiological status, as they are indicators for diuretic use, gastrointestinal losses, kidney disease, parathyroid condition, thyroid disease, cardiac failure etc¹⁷⁻¹⁹.”*

The AU and AEP solution used are commercialized by Biochemazone and the exact composition formula is proprietary. The relative concentration magnitude for K^+ , Na^+ and Ca^{2+} ions in AU is 10mM, 100mM and 1mM, respectively and that for all three ions in AEP is 1mM. We have included this information in Supplementary Table S2 and mentioned it in the main text as: *“The feasibility of the sensing system to operate in complex biofluid is demonstrated by testing the sensor behavior in artificial urine (AU) and artificial eccrine perspiration (AEP), where the relative concentration for K^+ , Na^+ and Ca^{2+} ions are within 1mM-100mM.”*

2. The abundance of sensors in the test is important for the improvement of test accuracy in this work. It’s really a perfect work to build such a highly integrated sensing platform, but workload and cost should also be balanced in a project. The authors should also discuss these two aspects in the paper. In addition, will the sensitivity/selectivity of the same sensor decrease after multiple tests? What about the shelf-life (although the sensitivity kept well in 10 h in Figure S7b)? How robust are the ISMs?

Our response:

We would like to thank the reviewer for this comment. The workload and cost of fabricating a highly integrated sensor chip is minimized by a wafer-scale process flow. Currently the sensor backbone is fabricated using a 4-inch wafer and results in 16 sensor chips in each wafer as shown in Fig. R1. The active sensing area takes only less than 10% of the current space therefore the footprint of the chip can be greatly reduced in the future. Photolithography is used which reduces cost while achieves higher throughputs. A double-layer photoresist structure is utilized to minimize photoresist residual on graphene. Wafer-scale graphene transfer is also commercially available. The cost for 8-inch monolayer graphene transfer is around \$800 for small volumes, thus the cost of one sensor chip (2cm×2cm) fabricated on an 8-inch wafer will be roughly under \$20, which is much cheaper than the commercialized glass ion-sensitive electrode (~\$250, *LabForce*). This work provides a promising way for future sensors, where both material cost and fabrication cost are in an affordable manner. With the reviewer’s comment, we added Fig. R1 as Supplementary Fig. 1c as well as the following discussion in Methods and Conclusion:

“In this work, we developed a highly integrated graphene-based sensing platform that overcomes the current limitations in 2D materials technology and achieves high performance and enhanced functionality. The scalable fabrication process provides a promising way for future sensors to achieve both excellent performance and low cost.”

“Graphene sensors were isolated from each other using oxygen plasma and a patterned PMGI/SPR700 resist stack as a mask. Photolithography is used to reduce cost while achieve

higher throughputs. ... The whole process can be easily scaled up for 6 or 8-inch process and the cost will be further reduced.”

As the reviewer mentioned, device stability is extremely important for reliable and accurate results. Here we discuss the device stability and robustness in three aspects:

- 1) **Measurement reproducibility:** Hysteresis, which will influence sensor accuracy and repeatability is minimized by optimized sweeping condition. The hysteresis present in electrolyte-gate graphene FETs is mostly dominated by capacitive gating and it can be suppressed by applying a slow sweeping rate as proved by Wang *et al.*^{R4}. We programed our sensing system to sample data at a rate of 20mV/s. As shown in Supplementary Fig. 5, the differences between two consecutive measurements are negligible under our sweeping condition.
- 2) **Stability over multiple measurements:** Sensor devices are measurement towards different ion concentrations for three times and the sensitivities are extracted and plotted in Fig. R2a (Fig. S11a). Sensitivities towards all three ions are stable over the 3 testing sessions. No significant sensitives degradation or drift were observed within multiple measurements.
- 3) **Stability over time:** Shown in Fig. R2b (Fig. S11b) is the sensor sensitivity with respect to storage time of 3 hours, 1 week, and 6 months. All three ISMs have stable sensitivity over time. The devices were stored in a nitrogen box at ambient temperature.

In summary, our sensor demonstrates reproducible performance over multiple measurements and long-term (at least 6 months) stable sensitivity towards K^+ , Na^+ and Ca^{2+} ions. To address reviewer’s comment, we have included the following figures in Supplementary Fig.1c, Supplementary Fig. 11 and covered it in the main text as “*The averaged sensitivity of ISM-functionalized sensor array is stable and repeatable over multiple measurements (Supplementary Fig. 11a). The robustness of the ion sensor array is further showcased by the negligible drift in sensitivity over a 6-month period (Supplementary Fig. 11b)*”

Figure R1: Image of 16 sensing chip fabricated on a 4-inch wafer.

Figure R2: Sensitivity of K^+ , Na^+ and Ca^{2+} ion sensors plotted **a** over multiple measurements; and **b** over long period of time (up to 6 months). All sensors show negligible drift and good stability.

3. Since there are more than 200 devices on the same sensor, the multi-channel measurement is important for the data collection. How many devices could be measured simultaneously in the testing process? How long does it take to finish the measurements on all the 215 devices in a test?

Our response:

We thank the reviewer for the question. The custom PCB measurement system is able to measure all sensors almost simultaneously. This is achieved by the 16-channel multiplexer on the board to switch quickly between different rows of the sensor array. For a standard V_{GS} sweep from -0.6V to 0.9V with 20mV/s sweeping speed, it takes 1.5 mins to finish the V_{GS} sweep for one sensor cell. With the current multiplexer, it roughly takes 3 minutes to finish the $I-V$ measurement of all 256 devices. We have added the following sentence in the main text to reflect reviewer's comment:

“The change in the source-drain current I_{DS} for each row and column combination of the sensor array are automatically measured as a function of gate-source and drain-source voltages, V_{GS} and V_{DS} . For a standard V_{GS} sweep (from -0.6V to 0.9V with 20mV/s sweeping speed), it takes roughly 3 minutes to finish measuring all 256 devices.”

4. According to the equation 3, the sensitivity is limited by Nernst limit (around 59 mV/decade). The sensitivity of some devices was actually higher than this value (shown in Figure 2a-c, Figure 3a, and Figure S7a). The authors should explain the deviations of sensitivity values, although some sensors with even higher sensitivity have been reported.

Our response:

We appreciate the reviewer's comment. The super-Nernstian behavior has been explored and demonstrated by many groups such as using defect-engineered graphene^{R5} and capacitive amplification via dual-gating^{R6}. In our devices, the super-Nernstian behavior arise mostly from uncontrolled defects sites due to material synthesis and device fabrication. If we consider the operation principle of the ISM, the membrane potential between ISM and electrolyte induces a potential at graphene/ISM interface and acts as an addition gate to the graphene channel^{R7}. In addition to the ideal electrostatic gating effect from the Nernst equation, defect sites on the graphene channel, including but not limited to grain boundaries, vacancies, contaminations, substrate doping etc., can induce charge transfer effect to the graphene channel. Such effect will modulate the channel fermi-level hence induces an additional leakage current to the gate and shifts in the I - V characteristics^{R5,8,9}.

The advantage of the highly integrated sensing system is the ability to use average performance to decrease the impact of device variations and leakage current, caused by immature material synthesis and fabrication process. In our system the sensors don't need to be perfect in order to provide adequate functionality.

We have added the following sentence in the main text and updated the **Supplementary Note 1** to discuss the super Nernstian sensitivity. *"Some devices show super-Nernstian behavior, which can be explained by uncontrolled charge transfer due to defect sites on graphene induced by fabrication process (Supplementary Note 1)."*

5. In Algorithm-enhanced sensing accuracy section, the authors should clarify if the multi-ion solutions used in the second set of experiments were in water or AU/AEP. It would be better if the authors could provide the sensing performance and classification results with real-world biological samples to support the practical significance of the sensing platform. Or the authors should at least discuss on how the sensing performance and data analysis could be affected due to the complexity of real-world biological samples.

Our response:

We appreciate the reviewer's comment. The multi-ion solution used in the model training was in water. We have clarified in the main text as: *"The second set, called "mixture solutions", contains electrolyte solutions with multiple ion types in deionized water and will be discussed in the following section."*

We added measurements of the as-fabricated sensor chip in real bio-fluid. Shown in Fig. R3a is the as-fabricated sensor array measured in water and in human serum. The sensing chip is fully functional, but a significant shift of the I - V characteristics is observed due to change in ionic strength and the presence of proteins and steroids. In addition, current level, cross-sensitivity and detection limit will also be different due to the interference from the complex environment. For a traditional sensor where only one type of sensor is present and absolute selectivity is assumed, the sensor readings with real-world biological samples might not be accurate. Our sensing system however introduced new approaches to tackle this problem, where a multiplexed sensor array was developed and combined with statistical/machine learning methods.

In order to have a practical functionality in clinical studies, the sensor system would need to be calibrated and trained with large sets of real biological samples to target towards specific

diseases. Such experiments are possible and will be included in future work, but the collection of real biological samples is outside of the scope of this paper, and it requires other advanced techniques to acquire precise ground truth labels. Instead, we trained another model to simulate how the system would be used in a real-world setting, where biofluids are analyzed using the concentration profile of the K^+ , Na^+ Ca^{2+} ions as biomarkers for electrolyte imbalance disorders. Here we re-categorized the mixture solution sets as shown in the Table R1 below and trained the model to predict the mixture concentration profile instead of the concentration of individual ions. We categorized the mixture solution sets following the relevant ion concentration in human plasma into four classes: Higher Calcium, Higher Potassium, Lower Sodium and Baseline.

The confusion matrix of the result model is shown in Fig. R3b (which is also added as Fig. 4f). We achieved an average accuracy of 84.7% for classification of ion concentration profiles with solutions containing multiple ion types. This analysis demonstrates the possibility of coupling the multiplexed sensing array with machine learning models to achieve fast and high accuracy of electrolyte imbalance-related disease diagnosis. The performance of the model could be greatly improved with a larger, balanced training dataset. The model could also be easily re-trained towards specific diseases with training data collected with real biofluid from patients in a clinic trial.

To address the reviewer's comment, we add the following discussion in the main text and SI:

“In additional to complex electrolyte profiles, biological fluids also contain antibodies, antigens, and hormones that introduce more confounders into the analysis. The graphene-based sensors will behave differently in terms of current level, Dirac Point, cross-sensitivity, detection limits as so on (Fig. 4f). The multiplexed sensor array analyzes the collective response of an array instead of assuming absolute sensitivity or selectivity of one device. Models can be readily calibrated or re-trained using real-world biological samples in future studies. Here we simulate how the system would be used in a real-world setting, where biofluids are analyzed using the concentration profile of the K^+ , Na^+ Ca^{2+} ions as biomarkers for electrolyte imbalance disorders. We re-categorized the mixture solution and trained the model to predict the mixture concentration profile instead of the concentration of individual ions (Supplementary Table 4). The confusion matrix of the result model is shown in Supplementary Fig. 24. We achieved an average accuracy of 84.7% for classification of ion concentration profiles with solutions containing multiple ion types. The model is less confident at identifying higher calcium and higher potassium. This could be due to the imbalanced training data in these classes, i.e., there are fewer observations with a higher calcium/potassium level that make the learning more difficult. This analysis demonstrates the possibility of coupling the multiplexed sensing array with machine learning models to achieve fast and high accuracy of electrolyte imbalance-related disease diagnoses. The performance of the model could be greatly improved with a larger, balanced training dataset. The model could also be readily re-trained towards specific diseases using training data collected with real biofluid from patients in a clinical trial.”

“Random Forest algorithm was used to demonstrate enhanced functionality of accurate ion type classification, concentration prediction, and potential applications in electrolyte imbalance-related disease diagnostics”

Figure R3: a) I - V characteristics of graphene sensor array tested in water (yellow curves) and in human serum (blue curves). b) confusion matrix for electrolyte imbalance classification in mixture solutions using integrated graphene sensing chip.

Table R1. Solution concentration of mixture solutions and label used for electrolyte imbalance classification model

Solution name	Ca ion concentration	K ion concentration	Na ion concentration	Label
Mix_Ca_1mM	1mM	5mM	30mM	Baseline
Mix_Ca_10mM	10mM	5mM	30mM	Higher Calcium
Mix_Ca_100mM	100mM	5mM	30mM	Higher Calcium
Mix_K_1mM	1mM	1mM	30mM	Baseline
Mix_K_10mM	1mM	10mM	30mM	Higher Potassium
Mix_K_100mM	1mM	100mM	30mM	Higher Potassium
Mix_Na_100μM	1mM	5mM	100μM	Lower Sodium
Mix_Na_1mM	1mM	5mM	1mM	Lower Sodium
Mix_Na_10mM	1mM	5mM	10mM	Lower Sodium
Mix_Na_100mM	1mM	5mM	100mM	Baseline

6. For the PCA, how many features were applied for the classification? The authors should clarify these features in the manuscript.

Our response:

We appreciate the reviewer's question. The number of features (dimensions) equals to the number of working devices on the sensing chip. PCA was used to reduce the dimensionality from >200 to 3 principal components. We performed two separate PCA analysis with

1) **Dirac point** as the feature value: The PCA matrix e has a dimension of $N \cdot M$ where N is the pure solution set with $N = \{K^{+}_{100\mu M}, K^{+}_{1mM}, K^{+}_{10M}, K^{+}_{100mM}, Na^{+}_{100\mu M}, Na^{+}_{1mM}, Na^{+}_{10M}, Na^{+}_{100mM}, Ca^{2+}_{100\mu M}, Ca^{2+}_{1mM}, Ca^{2+}_{10M}, Ca^{2+}_{100mM}\}$. The M features are the Dirac Point of individual devices on the sensing chip measured in each pure solution. The first and second principal component is plotted in Fig. R4a. The three types of ions are well separated with a total variance of 92.1%. This analysis demonstrates the multi-variate nature of the dataset produced by the highly integrated sensing chip with 3 ISMs.

2) **Sensitivity** towards K^{+} , Na^{+} , Ca^{2+} solutions of each individual device as features: The sensitivity is extrapolated as the slope of change in Dirac Point with respect to different concentrations. The PCA matrix has a dimension of $3 \times M$. The PCA is plotted in Fig. R4b and the first two principal components cover 100% of the total variance in the dataset. This result demonstrated that the sensitivity profile of the integrated sensing chip is significantly different and can be used to fingerprint ion types.

We modified the main text and added a paragraph explaining the details of PCA in the methods as: “We first performed Principal Component Analysis (PCA) using *the Dirac Point of individual devices as features* to visualize the multivariate data under a lower dimensional space while preserving the largest variance. *The first two principal components (PC) accounted for 92.1% of the total variance in the data. The score of PC1 and PC2 is plotted in Fig. 3e and the details of the PCA analysis are explained in Supplementary Note 7. The clusters of K^{+} , Na^{+} and Ca^{2+} ions are well-separated, indicating the sensor's ability to distinguish between different types of ions in electrolyte. Further separation can be achieved by using the ion sensitivities of individual sensors as the feature set as shown in Supplementary Fig. 20.*”

Figure R4: Principal component analysis (PCA) of integrated ion sensing chip towards electrolytes comprised of different of ions using **a** Dirac Points and **b** extrapolated sensitivity as feature vectors.

7. For the artificial urine (AU) and artificial eccrine perspiration (AEP), the authors may make their recipes opposite in Table S2 since there should be urea and uric acid in AU.

Our response:

We thank the reviewer for this comment. The artificial urine (AU) and artificial eccrine perspiration (AEP) solution are commercialized product by Biochemazone™. The AU solution does contain Urea and Uric acid. Comparing to AEP, AU has more complex anion environment and more metabolites. Our sensor has demonstrated stable operation in both solution environments as shown in Supplementary Fig 13. Following reviewer’s comment, we updated **Supplementary Table S2** as shown below to better highlight the differences between AU and AEP.

Table R2. Molecule contents of Biochemazone™ artificial urine (AU) and artificial eccrine perspiration (AEP).

		artificial eccrine perspiration (AEP)	artificial urine (AU)
Cation	K ⁺	Y	Y
	Na ⁺	Y	Y
	Ca ²⁺	Y	Y
	Mg ²⁺	Y	Y
	NH ₄ ⁺	Y	Y
Anion	Cl ⁻	Y	Y
	PO ₄ ³⁻		Y
	C ₆ H ₅ O ₇ ³⁻ (Citrate ion)		Y
	SO ₄ ²⁻	Y	Y
	SO ₃ ²⁻		Y
	HPO ₄ ²⁻		Y
	H ₂ PO ₄ ⁻		Y
Metabolites	Uric Acid	Y	Y
	Urea	Y	Y
	Creatinine		Y
	Lactic Acid	Y	
	Acetic Acid	Y	
	Histidine	Y	

8. As shown in Figure 2d-f, the sensitivity changed with the opposite test sequence (from high concentration to low concentration). Could it be explained with the cross contamination? At the same time, there are only three data points (red one) in Figure 2d-f. What about the relative conductance changes at other two concentrations? Lastly, only $\Delta G/G$ was applied for the

investigation of reversibility in the manuscript, the Dirac point change from high concentration to low concentration should also be plotted.

Our response:

We thank the reviewer for this comment. Cross contamination is possible especially for measurements going from high concentration to low concentration, but the effect has been mitigated by the rinsing step while switching solutions. Fig. R5a shows the reversibility plot of Na ISM measured to 10 μM and similar reversibility was obtained. Five orders of magnitude change in ion concentration were measured in order to identify the detection range of the sensor system. Since only three points are enough to extrapolate the backward slope, most of the reversibility tests only measured up to 100 μM in the manuscript.

The reversibility of our ion sensors originated from its sensing mechanism, where the Nernstian potential drop between membrane and electrolyte is determined by the ion concentration in the electrolyte. Decrease in ion concentration will induce a righthand shift of the I - V characteristics and the Dirac Point will change from low to high. We have also performed the full I - V measurements for the Na ISM functionalized sensor reversibility as an example and the results are shown in Fig. R5b. Right-hand shift and increase in Dirac point values were. The sensor's reversibility is quantified by calculating the percentage difference between slopes fitted with forward measurements and that fitted with backward measurements. The insert show extracted Dirac point reversibility of individual devices on the sensing chip and highlights that the reversibility can be improved significantly by averaging over a large number of devices.

This paper is mainly aimed at showing the advantage of using average performance to decrease the impact of the device variation caused by immature material synthesis and fabrication process, rather than to report a sensor with perfect selectivity and reversibility. In our system the sensors don't need to be perfect in order to provide adequate functionality.

We have included the following figures as **Supplementary Fig. 14** and mentioned in the main text as *“The average difference of fitted slopes is below 10% while that of the worst case of an individual device could be over 80%. Similar reversibility results were observed using the shift in Dirac Point instead of channel conductance (Supplementary Fig. 14)”*

Figure R5: The average change in **a** channel conductance and **b** Dirac Point for Na^+ ISM functionalized sensing chip showing excellent reversibility over several magnitude change in Na^+ concentration

9. The authors claimed that only one reference solution was needed for the determination of ions in a unknown sample with the profile-matching calibration. Is a reference solution always essential for the calibration in each test, or not needed any more after the calibration of sensor with the reference? In addition, what was the concentration of reference for the calibration in Figure 2i? Should the concentration of reference always fall in the linear range?

Our response:

We thank the reviewer for this comment. Calibration is essential for almost all sensing systems in order to have precision and reproducibility^{R10}. Sensor accuracy can deteriorate through wear, aging and environmental influences and should therefore be recalibrated at regular intervals. Similarly, our ion sensor arrays will also need to be re-calibrated due to unavoidable drift over time due to graphene's property change in ambient air. Shown in Fig. R6 is the extracted Dirac Point of a K^+ ISM functionalized array for tests within the same day and tests within a week. Negligible drift was observed for the two tests in the same day. Noticeable drift was present for the measurement after a week, where a re-calibration was necessary for accurate measurement. A general rule-of-thumb for accurate results is to re-calibrate if measurements are not done in the same day. Traditionally such re-calibration would require multiple calibration solutions. With the profile-matching technique proposed in the manuscript however, only one calibration solution is needed as long as the sensitivity is stable over long period of time. We have showed the stability of sensitivity for up to 6 months as shown in Fig. R2b.

Concentration of reference solution indeed need to be within the linear range because we are utilizing the quasi-linearity in graphene's I - V curves. The linear range for our ion sensors is wide enough (5 order of magnitude) for most applications. The reference concentration used in the manuscript is 1mM. The following modification is made to the main text and supplementary information:

“Calibration is essential for almost all sensing systems since sensors can deteriorate through wear, aging and environmental influences³⁵. Similarly, electrolyte-gated graphene field-effect transistors (EGFETs) will also need to be re-calibrated due to the unavoidably drift over time resulting from graphene’s property change in ambient air (Supplementary Fig. 15). Typical methods for calibrating graphene EGFETs perform full I-V characterization of the devices under multiple dilutions spanning the entire range of interest³⁶⁻³⁸. The profile-matching method however requires only a single measurement of I-V characteristics using one calibration solution of known concentration. The calibration solution should be within the linear range of the device (10 μ M to 100mM) in order to utilize the quasi-linearity in graphene’s I-V curves. Since the sensitivity drift of the ion sensors is negligible over six months as shown in Supplementary Fig. 11, the sensors can be easily calibrated with the profile-matching approach to achieve same sensing accuracy over multiple testing sessions.”

Figure R6: Dirac Point drift of K ISM functionalized sensors over a week showing the necessity of fast and easily calibration schemes.

10. In Figure 4f, what was the concentration of each ion for the test? Was the mixture solution used for the test?

Our response:

We appreciate the reviewer’s question. Fig. 4f (Fig. 4g after revision) is plotting the impact of sensor redundancy in model accuracy. The pure solution set was used for this analysis. The ion concentrations for the pure solution set are listed below as Table R3 and in the Supplementary Information as Table S3. Mixture solution was not used in this analysis. In order to predict the ion concentration in a mixture solution, we need to run all three ion models. Therefore, it is not reasonable to separately compare each ion model’s accuracy with mixture solutions. We added the following sentence in the main text to clarify: “Fig. 4g is the accuracy of different tasks sampling from 1, 1/2, 1/4 and 1/8 of the total devices using the pure solution set”.

Table R3. Solution concentration for sensor testing

Solution name	Ca ion concentration	K ion concentration	Na ion concentration
Pure_Ca_1μm	1μm	0	0
Pure_Ca_10μm	10μm	0	0
Pure_Ca_100μm	100μm	0	0
Pure_Ca_1mm	1mm	0	0
Pure_K_10um	0	10um	0
Pure_K_100um	0	100um	0
Pure_K_1mm	0	1mm	0
Pure_K_10mm	0	10mm	0
Pure_K_100mm	0	100mm	0
Pure_Na_1μm	0	0	1μm
Pure_Na_10μm	0	0	10μm
Pure_Na_100um	0	0	100μm
Pure_Na_1mm	0	0	1mm
Pure_Na_10mm	0	0	10mm
Pure_Na_100mm	0	0	100mm

11. For the classification of different ion concentrations in the mixture solution, the confusion matrix is still limited to each ion, which contains five classes. Could the authors provide the confusion matrix with fifteen classes? (Three types of ions and five different concentrations).

Our response:

We appreciate the reviewer's comment. For the mixture solution, three independent models were trained to give predictions for K^+ , Na^+ and Ca^{2+} ion concentration therefore three separate confusion matrices were reported. If we train one model to predict the concentration of all three ions in a mixture, the confusion matrix dimension would be $5 \times 5 \times 5$, assuming five different concentrations for each ion. The total labeled class will be 125. This approach is possible if we have larger, much diverse training data, but it is not feasible for our current data size.

12. The errors in units should be corrected. "mm" should be "mM" in page 15, 18 and 30.

Our response:

We appreciate the reviewer's feedback. We have carefully checked the manuscript and corrected the units.

Reply to reviewer 2's remarks:

Overall remarks:

Xue et al. reported a field-effect sensing platform based on more than 200 integrated graphene ion sensors. The authors claimed that they overcome the limitation of relatively large device-to-device variation in 2D materials, and achieved high-performance ion sensing by adopting new calibration methods as well as Random Forest algorithm. The manuscript is well-written and the technique seems to be reasonable. Given the research tackles one of the most important limitations/problems in biochemical sensors based on non-uniform 2D materials, in my opinion this paper should be accepted with minor revision.

Our response:

We thank the reviewer for the positive and encouraging comments.

1. In Supplementary Fig. 1f, the performance of the device is quite different. I suggest giving statistical data on resistance or carrier mobility for the same batch of devices for comparison. There are only few characterizations on the surface morphology and quality of the graphene devices. The authors should show optical images of individual devices (as-fabricated and functionalized).

Our response:

We appreciate the reviewer's comment and suggestions. Here we provide more statistical data on the device variation of as-fabricated sensors. Fig. R7a insert shows the histogram of the extracted Dirac Point of an as-fabricated sensor array. The average Dirac Point for this batch lies at a 97.1 mV with a standard deviation of 40.7 mV. We also performed I_{DS} - V_{DS} characterization of the devices after fabrication as shown in Fig. R7b. The sensor chip is measured in water with $V_{GS} = 0$ and V_{DS} sweep from 0 to 0.5 V. Shown in Fig. R7d is the extracted channel resistance at different V_{GS} bias. The range of channel resistance varies with different biasing condition. Fig. R7c is a color map of channel resistance for the whole array with $V_{DS} = 250$ mV and $V_{GS} = 0$ V. We have added the following figures in supplementary information as **Supplementary Fig. 3** and mentioned them in the main text as:

“The drain current with respect to gate voltage (will be referred as “I-V”) characteristics as well as the drain current with respect to drain voltage characteristics of the sensor chip before functionalization are shown in Supplementary Fig. 3a-b...Large device-to-device variations in terms of current level, channel resistance, Dirac Point, (the location of the minimum conduction point), and shape of the I-V characteristics are present for the >200 working devices on one sensing chip (Supplementary Fig. 3)”

The as-fabricated devices have different performance, which could be from material intrinsic variations or fabrication induced variability. It would be problematic for a traditional

sensor, where only one or two devices are present, if the batch-to-batch variation is a magnitude different. However, with our highly integrated sensing system, we are averaging out the device-to-device variation and only look at the universal trend of the 200 devices on one chip by applying statistical analysis and machine learning algorithms. In this case the device-to-device variation in the I - V curves are not an issue for our sensing system.

In addition, we also added the Fig. R8, optical image of as-fabricated graphene sensor array and functionalized sensor array, as Supplementary Fig. 1d-e and mentioned them in the main text as:

“Each sensing unit consists of a $30 \times 30 \mu\text{m}$ graphene channel with two Ti (5 nm)/Au (150 nm) source/drain electrodes. The optical image of as-fabricated graphene sensor arrays is shown in Supplementary Fig. 1d.”

“Details of printing recipe can be found in Methods and the optical image of the sensor array after ISM printing is shown in Supplementary Fig. 1e.”

Figure R7: Variation in as-fabricated graphene sensor array. **a** I - V characteristics of an as-fabricated graphene sensing chip. The sensing chip was tested in water with an Ag/AgCl reference

electrode and biased at $V_{DS} = 250$ mV. Insert shows the histogram of Dirac points extracted from **a** with an average Dirac Point of a 97.1 mV \pm 40.7 mV; **b** I_{DS} - V_{DS} characteristics of an as-fabricated graphene sensing chip. Sensor chip is tested in water with zero gate bias; **c** color map of channel resistance for the whole array with $V_{DS} = 250$ mV and $V_{GS} = 0$ V; **d** graphene channel resistance distribution under various gating conditions.

Figure R8: Optical image of **a** an as-fabricated graphene sensor array. White rectangular box in the inserts is the outline of a graphene channel. **b** integrated sensor array with Ca, Na and K ISM printed on the separate area.

2. Material jet printers are used to deposit various chemicals onto the sensing area with precise lateral control. To what extent the printers can control the uniformity of the functionalization?

Our response:

We would like to thank the reviewer for this comment. Here we would like to discuss a few parameters that have an impact on membrane quality or device performance:

1. Thickness: The membrane thickness can be controlled by a couple printing parameters including but not limited to nozzle size, pressure, and dispense pulse time. Here we characterize the uniformity of thickness by printing the PVC membrane into a 2.5-by-0.5 mm strip with the 50 μm nozzle at various pressure. Five measurements were taken with a digital caliper across the membrane strip and the averaged thickness is plotted in Fig. R9. With the other printing parameters unchanged, higher pressure results in a thicker membrane. The standard deviation of the membrane thickness is 0.01 mm, which is close to the precision of the caliper. In fact, membrane thickness has no direct effect on sensitivity or response time but has some effect on detection limit. Thinner membranes could increase the transmembrane ion flux and worsen the detection limit. We have demonstrated near perfect sensitivities for all three ions down to 10 μM. The lower bound concentration for K^+ , Na^+ , Ca^{2+} in physiological biofluid such as urine, sweat and blood are generally in the mM range. Therefore, our ISM membrane thickness and detection limit are more than sufficient as a bioelectronic for ion sensing and can even be used in some more delicate situations that have even lower ion concentrations such as intracellular ion sensing.

2. Roughness: roughness also has minimal effect on sensitivity but might decrease the effective charge-carrier mobility in graphene. Supplementary Fig. 17 shows the AFM mappings of $5\ \mu\text{m} \times 5\ \mu\text{m}$ ISM/graphene areas. The average R_q for all three membranes is less than 4 nm and average R_a is less than 3 nm. We expect non-significant effect on device sensitivity, detection range and stability with such surface small roughness on the membrane. We also extracted the carrier mobility of graphene with the present of ISM and the average is above $500\ \text{cm}^2/\text{V}\cdot\text{s}$, which is adequate for electrolyte-gated graphene transistors reported in literature.

This paper is mainly aiming to provide insights on how to turn average performing, or even poor performing transistors with average material quality into a highly functional and powerful sensing platform. Device variation that might be caused by nonuniform functionalization deposition would be an issue for traditional sensors where only a couple devices were used. By averaging over >200 devices on the sensor array, we have shown significantly increased sensitivity, reversibility, as well as accuracy as shown in Fig. 2 a-f and Fig. 4g.

To address reviewer's comment, we have included Fig. R9 as **Supplementary Fig. 18** and covered it in the Methods as *"We adjusted the applied pressure to 20-30 psi and the stroke to 60% - 70% (percentage of voltage drop during the valve actuation) with the closed valve voltage of 100 V. The membrane uniformity over an area of $2.5\ \text{mm} \times 0.5\ \text{mm}$ is characterized and shown in Supplementary Fig. 18.*

Figure R9: Average membrane thickness over an area of $2.5\ \text{mm} \times 0.5\ \text{mm}$ printed by the material jetting printer with different printing pressure. Five measurements were taken with a digital caliper across each membrane strip.

3. Fig. 3c gives the ion-selective results for ion-selective membranes with different membrane functionalizations, where Ca ISM exhibits a non-selective sensing response towards Na^+ ions. Did the authors test the ionic sensing response of the as-fabricated graphene devices? It is worth noting that previous works (Nano Lett. 2011, 11, 3597 and Adv. Mater. 2017, 29, 1603610) demonstrated

that intrinsic defects, as well as possible contaminations introduced during device fabrication and storage, might lead to uncontrolled sensitivity.

Our response:

We appreciate the reviewer’s comment. One of the advantages of the highly integrated sensing system is the ability to use average performance to decrease the impact of the device variation caused by immature material synthesis and fabrication process. In addition, we demonstrated that imperfections and variability in novel materials can be used to improve system level performance and calibration.

We acknowledge that intrinsic defects in advanced 2D materials and contaminations introduced during fabrication will impact device performance in terms of electrochemical sensitivities, mobility, contact resistance, hysteresis and so on^{R11}. It would be problematic for a traditional sensor, where only one or two devices are present, if the batch-to-batch variation is significantly different. However, with our highly integrated sensing system, we are averaging out the device-to-device variation and only look at the universal trend of the >200 devices on one chip by applying statistical analysis and machine learning algorithms. In this case the device-to-device variation in the I - V curves is not an issue for our sensing system.

We measured the sensitivity of as-fabricated graphene sensors towards K^+ , Na^+ , Ca^{2+} as shown in Fig. R10a. Defects introduced from device fabrication could lead to uncontrolled sensitivity if not functionalized as indicated by I - V shift in Fig. R10b and the large error bar in Fig. R10a. But averaged data of bare graphene shows in-significant sensitivity towards ions. Ion sensitivity and selectivity of ISM functionalized graphene sensors are more stable and controlled due to protection of PVC membranes and the overpowering Nernstian effect. Averaging over many devices further eliminate uncontrolled sensitivity that could originated from defects and contaminations. To address this comment, we added the following discussion in the main text “*The intrinsic sensitivity of bare graphene towards K^+ , Na^+ and Ca^{2+} is also characterized. Although some of the individual sensors show sensitivity towards change in ion concentration due to defects on the graphene channels³¹, the averaged data shows negligible sensitivity (Supplementary Fig. 10). This further demonstrated the effectiveness of averaging over many devices to eliminate uncontrolled sensitivity that could originated from defects and contaminations.*”

Figure R10: Responses of as-fabricated graphene sensors. **a** average Dirac Points shift with respect to changes in K^+ , Na^+ and Ca^{2+} ion concentration. Error bar is the standard deviation over multiple devices. **b** shift of I - V characteristics of an as-fabricated graphene sensor without functionalization towards changes in Na^+ ion concentration.

4. In my opinion, a reliable and stable operation of the graphene devices is the key. The authors should provide long-term stability measurements of as-fabricated and functionalized graphene sensors.

Our response:

We appreciated the reviewer's comment and agreed that stability is the key for robust and reliable measurement. Here we provide the long-term stability of the sensor chip for up to 6 months. Shown in Fig. R2b is the sensor sensitivity with respect to storage time of 3 hours, 1 week, and 6 months. The error bar indicates the sensitivity standard deviation of all the working devices on the chip. The sensor chips are stored in a nitrogen box at ambient environment. The sensitivities of functionalized graphene sensors show negligible drift over half a year. We benchmarked the ion sensors' long-term stability with counterparts in literature and industry as listed in Table R4. The long-term stability of our graphene ion sensor arrays is longer than most of the sensors reported in the literature and comparable to commercialized ISEs.

In addition to long-term stability, we also demonstrated the reproducible sensitivity between multiple measurements as shown in Fig. R2a. Following reviewer's comment, we added the updated Supplementary Table S1 and Fig. R2 as Supplementary Fig. 11 and added the discussion: "*The averaged sensitivity of ISM-functionalized sensor array is stable and repeatable over multiple measurements (Supplementary Fig. 11a). The robustness of the ion sensor array is further showcased by the negligible drift in sensitivity over a 6-month period (Supplementary Fig. 11b).*"

"Comparing to other reported ion sensors, which normally have less than five devices tested or reported, our ion sensing arrays exhibit excellent sensitivity, good reversibility, large detection range and long-term stability (Supplementary Table S1)."

Figure R2: Sensitivity of K⁺, Na⁺ and Ca²⁺ ion sensors plotted **a** over multiple measurements; and **b** over long period of time (up to 6 months). All sensors show negligible drift and good stability.

Table R4. Performance comparison of ion sensors-based ion-selective membranes

Device type	Materials	Target Ion	Detection Range	Sensitivity [mV/decade]	# of device tested	Stability
ISE	PEDOT:PSS/Au	Ca ²⁺	0.25 mM - 2 mM	32.7±0.981 ^[S4]	6	90 mins of continuous measurements
ISE	PEDOT:PSS/Au	Ca ²⁺	1 mM - 10 mM	18.3±1.7 ^[S5]	8	2 days of continuous measurements
ISE	Ag/ZnO	Ca ²⁺	100 nM - 10 mM	29.67 ^[18]	-	N/A
ISE	Graphene/Carbon glass	K ⁺	30 μM - 100 mM	59.2 ^[S7]	-	3 weeks
ISE	Au	Na ⁺ , K ⁺	10 - 160 mM, 1 - 32 mM	64.2, 61.3 ^[S8]	8	5 weeks
ISFET	Graphene/Au	K ⁺	10 nM - 1 mM	7.8 ^[S9]	1	N/A
Chem-resistor	Self-assembled graphene/Au	Na ⁺ , K ⁺ , Ca ²⁺ , H ⁺	2 - 5 mM	- ^[21]	1	N/A
ISFET	Graphene/Au	K ⁺ , Na ⁺ , Cl ⁻ , etc.	10 μM - 100 mM	49.2, 45.7, -43.0 ± 0.2 ^[6]	4	5 months
Chem-resistor	MoS ₂ /Au	Na ⁺ , Pb ²⁺ , Hg ²⁺ , Cd ²⁺	-	- ^[22]	5	N/A
ISE	LabForce Commercial device	K ⁺	1nM - 1M	56±3	1	12 months
ISE	LabForce Commercial device	Na ⁺	4.4nM - 1M	55±3	1	12 months
ISE	LabForce Commercial device	Ca ²⁺	500nM - 1M	26±2	2	12 months
ISFET (this work)	Graphene/Au	K ⁺ , Na ⁺ , Ca ²⁺	10 μM - 100 mM	-54.7 ± 2.90, -56.8 ± 5.87, -30.1 ± 1.90	>200	6 months

Reply to reviewer 3's remarks:

Overall Remarks:

In this manuscript, the authors present a novel approach to address current challenges in 2D material-based sensing devices such as material quality variability, device uniformity, high performance and enhanced functionality in ion classification application. The proposed approach reduces the requirements on material quality and device uniformity. To this end, the authors fabricated sensor arrays consisting of 16×16 graphene devices to provide more than 200 working sensing units for each chip, and configured them to achieve multi-ion sensors by modifying their surface with three different ion-selective membranes (ISMs). Moreover, machine learning algorithms such as the Random Forest algorithm, were applied to demonstrate the enhanced functionality of accurate ion type classification and concentration prediction. The developed platform demonstrates near ideal sensitivity, excellent reversibility, and a large detection range for each type of sensor despite the large non-uniformity of the individual devices. The sensing platform could be readily adopted for other analytes of interest as well as with other advanced 2D materials that also suffer from the same issues (e.g., device-to-device variations) to realize accurate and reliable multiplexed sensing in biomedical applications.

The manuscript is interesting and generally fulfils the scope of Nature Communications. However, in its present form it can not be recommended for publication. To provide an adequate contribution to the journal, the authors have to take into account the following points in a major revision.

Our response:

We would like to thank the reviewer for the positive comments and all the constructive suggestions/comments. We have

- 1) perform additional experiments on device variations, model benchmark, and real bio-fluid measurement to showcase the robustness and effectiveness of the sensing system
- 2) provide detailed discussion on device filtering methods and principal component analysis
- 3) revised the manuscript to fix the typos and to further polish the discussion

The detailed responses to technical questions the reviewer raised in detail below.

1. On page 3, the full name of the abbreviations "PEDOT" and "PSS" should be given at their first appearance in the manuscript.

Our response:

We thank the reviewer for the comment. We have modify the main text accordingly as *"The high carrier mobility, which translates into high transconductance, makes graphene a more desirable transducer comparing to organic materials such as poly(3,4-ethylenedioxythiophene) polystyrene sulfonate (PEDOT:PSS)."*

2. On pages 3 and 4, the two paragraphs, "Here we demonstrate a novel " and "In this paper, we fabricate arrays of" are not well organized thus hard to understand. There are a lot of redundant elements, such as the sentence "Instead of focusing on the improvement of intrinsic material quality, fabrication uniformity and surface functionalization, we demonstrate ways to overcome the large degree of variability of advanced materials by developing a high-density graphene-based sensor array, thus significantly reducing the requirements on material quality and device uniformity. ". The section following "thus" is repeated as the section following "Instead of", etc.

Our response:

We thank the reviewer for the constructive feedback. We have re-organized and shorten the paragraphs mentioned. The revised text follows the structure of 1) overcome variations with high-density graphene-based sensor array; 2) utilize variations with statistical methods and machine learning algorithms.

Here is the revised paragraph following reviewer's feedback:

"Here we demonstrate a novel approach to overcome the challenges in 2D material-based sensing devices and achieve high performance and enhanced functionality. Rather than focusing on the improvement of intrinsic material quality, fabrication uniformity and surface functionalization, we develop a high-density graphene-based sensor array platform to overcome the large degree of variability of advanced materials. We fabricate arrays (16×16) of graphene devices to provide more than 200 working sensing units for each chip, and configure them into multi-ion sensors by functionalizing their surface with three different ion-selective membranes (ISMs) We demonstrate near-ideal sensitivity, excellent reversibility, and large detection range for each type of sensor despite non-uniformity in individual devices. The variations and imperfections in material synthesis and device fabrication can be leveraged by statistical analysis and machine learning algorithms. A profile-matching calibration method utilizing sensor non-uniformity and redundancy is introduced to eliminate the need for multiple calibration solutions, which is especially useful for sensing applications targeting portability and field use. A random forest algorithm is used to quantify analyte concentrations in the presence of multi-ions. The abundance (N>200) and multiplexity of sensors and sensor types are shown beneficial for improving model accuracy. We demonstrate that system-level co-design of sensing arrays and algorithms significantly improves sensor performance thus enabling rapid prototyping and in-depth data analysis in spite of the limitations present in graphene and other advanced 2D materials."

3. The authors present the Raman characterization before and after the sensor fabrication. Did the authors also perform the Raman mapping characterization for the graphene film after sensor fabrication procedure? This could be helpful to show the quality of whole graphene film after fabrication.

Our response:

We appreciate the reviewer's comment. Raman mapping characterization shown in Supplementary Fig. 2b is 2D/G ratio mapping of a 10 mm ×10 mm graphene film before fabrication. This result shows that the graphene film is single layer. After fabrication, the graphene

film is isolated into 256 mesas which form the channels for the transistor array. The same mapping characterization is not applicable anymore since graphene is no longer a complete $10\text{ mm} \times 10\text{ mm}$ sheet. Moreover, process induced damage on graphene's quality is typically better reflected onto the D band^{R12,13}. Therefore, we perform a D/G Raman mapping of $20\text{ }\mu\text{m} \times 20\text{ }\mu\text{m}$ on a graphene channel after all the fabrication procedure. The mapping is shown in Fig. R11a. Variations of I_{D_band}/I_{G_band} value is noticeable, but the average ratio is below 0.15, indicating minimal defects on the graphene channels. We also sampled 20 channels on the as-fabricated sensor array and performed point Raman characterization to compare the D band intensity variation across the whole array (Fig. R11b). This result shows that the fabrication introduced minimal damage on the graphene film.

However, it is still unavoidable to have non-uniformity and localized defects or tears as indicated by the outliers in Fig. R11a and the I - V characteristics of as-fabricated graphene sensors in Supplementary Fig. 3a. In the main text we have shown that by having a large sensor array, we are able to not rely on the quality of the graphene while still have high performance sensor system. We added the figures as **Supplementary Fig. 2** and the following changes was made to the Methods: *“The Raman Spectrum after fabrication is shown in Supplementary Fig. 2c and the D/G band mapping of a $20\mu\text{m}$ -by- $20\mu\text{m}$ graphene channel is shown in Supplementary Fig. 2d. Insert in Supplementary Fig. 2c shows the distribution of I_{D_band}/I_{G_band} ratio sampled from 20 devices across the array. The low D band intensity indicates low density defects and minimal damage to the graphene channel.*

Figure R11: **a** I_{D_band}/I_{G_band} ratio map of a $20\text{ }\mu\text{m} \times 20\text{ }\mu\text{m}$ graphene channel after fabrication. The average of the of I_{D_band}/I_{G_band} ratio is 0.129 with a standard deviation of 0.097. **b** Histogram showing the distribution of the of I_{D_band}/I_{G_band} ratio of graphene channels on the sensing array after fabrication. 20 devices were sampled on the same array.

4. In the manuscript, the authors point out that there are $16 \times 16 = 256$ sensors in the array. However, only 215 working devices were functionalized and presented, as shown on page 7. The authors should explain why not the 256 sensors are utilized and ways to overcome this limitation.

Our response:

We would like to thank the reviewer for this comment. The sensor yield is generally above 80%. The non-functioning devices on the sensor array are filtered out before data analysis. The filtering conditions are the following:

- Filter out possible shorted channel: $I_{DS} > 170 \mu A$
- Filter out possible broken channel: $I_{DS} < 2 \mu A$
- Filter out abnormal channel:
 - Dirac Point not within the V_{GS} sweeping range (-0.6 V – 0.9 V)
 - $1 < I_{DS,max} / I_{DS,min} < 10$

As demonstrated in the main text, sensor redundancy can greatly enhance the overall accuracy (Fig. 4g and Supplementary Fig. 25). A higher yield means more working devices to analyze which could lead to better performance in complicated tasks such as ion concentration prediction. The yield can be improved by advances in material synthesis, transfer methods as well as optimizing fabrication process. The device filtering criteria is added to the supplementary information and mentioned in the main text as: *“Non-functional pixels were filtered out using the criteria outlined in Supplementary Note 2 and the average yield for the sensing chip is >80%.”*

5. In the section "Highly integrated Array for Multiplexed Sensing": The authors should comments on the observation "Na membrane shows a more neutral functionalization". Which are the underlying mechanisms?

Our response:

We think that one of the reasons that contribute to the more neutral functionalization effect of the Na ISM is the molecular structure of the Na ISM (Fig. R12b). It consists of a calix[4]arene structure that prefers to stay in the “cone” conformation where the ester groups are on the same side^{R14}. Moreover, calixarenes are believed to interact with graphene surface via hydrophobic interaction and π - π stacking^{R15}. This would lead to the polar functional group in most of the Na ISM molecules pointing away from the graphene surface, hence resulting in the more neutral functionalization observed. On the other hand, K and Ca ISMs (Fig. R12a and Fig. R12c) have flatter and more flexible molecular structures that would result in a closer distance between the functional group and the graphene surface, resulting in a more significant doping effect. We have added the molecular structure of the ionophores as Supplementary Fig.19 and the discussion as Supplementary Note 6 and mentioned in the main text as: *“The difference in the doping effect can be attributed to the molecular structure of different ionophores^{40,41} (Supplementary Note 6).”*

Figure R12: Molecular structure of the ionophore for **a** K ISM, **b** Na ISM and **c** Ca ISM

6. In Figure 3(d), the author show the PCA score plot of the chip sensitivity, but the introduction of the input features (multivariate) are not explained in the manuscript. This should be improved.

Our response:

We appreciate the reviewer’s question. The number of features (dimensions) equals to the number of working devices on the sensing chip. PCA was used to reduce the dimensionality from >200 to 3 principal components. We performed two separate PCA analysis with

1) **Dirac point** as the feature value: The PCA matrix e has a dimension of $N \cdot M$ where N is the pure solution set with $N = \{K^+_{100\mu M}, K^+_{1mM}, K^+_{10M}, K^+_{100mM}, Na^+_{100\mu M}, Na^+_{1mM}, Na^+_{10M}, Na^+_{100mM}, Ca^{2+}_{100\mu M}, Ca^{2+}_{1mM}, Ca^{2+}_{10M}, Ca^{2+}_{100mM}\}$. The M features are the Dirac Point of individual devices on the sensing chip measured in each pure solution. The first and second principal component is plotted in Fig. R4a. The three types of ions are well separated with a total variance of 92.1%. This analysis demonstrates the multi-varitative nature of the dataset produced by the highly integrated sensing chip with 3 ISMs.

2) **Sensitivity** towards K^+ , Na^+ , Ca^{2+} solutions of each individual device as features: The sensitivity is extrapolated as the slope of change in Dirac Point with respect to different concentrations. The PCA matrix has a dimension of $3 \times M$. The PCA is plotted in Fig. R4b and the first two principal components cover 100% of the total variance in the dataset. This result demonstrated that the sensitivity profile of the integrated sensing chip is significantly different and can be used to fingerprint ion types.

We modified the main text and added a paragraph explaining the details of PCA in the methods as: “We first performed Principal Component Analysis (PCA) using *the Dirac Point of individual devices as features* to visualize the multivariate data under a lower dimensional space while preserving the largest variance. *The first two principal components (PC) accounted for 92.1% of the total variance in the data.* *The score of PC1 and PC2 is plotted in Fig. 3e and the details of the PCA analysis are explained in Supplementary Note 7.* *The clusters of K^+ , Na^+ and Ca^{2+} ions are well-separated, indicating the sensor’s ability to distinguish between different types of ions in electrolyte.* *Further separation can be achieved by using the ion sensitivities of individual sensors as the feature set as shown in Supplementary Fig. 20.”*

**Figure R4**: Principal component analysis (PCA) of integrated ion sensing chip towards electrolytes comprised of different of ions using **a** Dirac Points and **b** extrapolated sensitivity as feature vectors.

7. In the "Algorithm-enhanced sensing accuracy" section, the authors present the classification accuracy implemented by the random forest algorithm. Did the authors perform the classification with other classifier algorithms, such as LDA, SVM, KNN, etc? Which results can be expected?

**Our response:**

We would like to thank the reviewer for this constructive comment. We trained the models with different classification algorithms for the algorithm performance comparison. Shown in Fig. R13a is the model accuracy with Latent Dirichlet Allocation (LAD), Support Vector Machine (SVM), K-Nearest Neighbors (KNN), Gaussian Process (GP) and Random Forest (RF). Note five iterations were performed on each model and the average accuracy is plotted and the standard deviation of the model accuracies is within 5%. The model implemented by the Random Forest algorithm has comparable or better accuracy compared to other classification algorithms. The training time is also comparable (<5s) for all algorithms except for GP, which is significantly slower (>170s) (Fig. R13b). This result shows that the multi-dimensional dataset gathered from the graphene-based sensor arrays are compatible for various machine learning algorithms. We also attempted to use deep learning methods such as neural networks. However, such methods require large amounts of data in order to outperform other algorithms.

We added the following discussion to the main text to address this comment: "*Due to its tree-based nature, random forest is also much more interpretable and less data-hungry than complex learning frameworks such as deep neural networks, making it more suitable for the dataset from our sensing chip. Other classification algorithms that are commonly used for disease prediction problems[40], including Latent Dirichlet Allocation (LAD), Support Vector Machine (SVM), K-Nearest Neighbors (KNN) and Gaussian Process (GP), are also compatible with the dataset gathered by the multiplexed sensing chip as shown in Supplementary Fig. 23. The model implemented by the Random Forest algorithm has comparable or better performance compared to other classification algorithms.*"

Figure R13: Comparison of **a** average accuracy and **b** average training time for models trained with Latent Dirichlet Allocation (LAD), Support Vector Machine (SVM), K-Nearest Neighbors (KNN), Gaussian Process (GP) and Random Forest (RF).

8. On page 15, the authors conclude the accuracies of 1/8 chip is larger than most of the reported sensors. The authors should give more detailed results of the other works and include the necessary citations.

Our response:

We thank the reviewer for this comment. Very few papers in the literature have demonstrated multi-ion sensing in mixture solution. Fakih *et al.* reported an accuracy of ± 0.01 log concentration units for Na^+ , K^+ and NH_4^+ ions in mixture solution by correcting the sensor responses with Nikolskii–Eisenman formalism^{R16}. Hanitra *et al.* attempted to simulate the behavior of polymeric solid-contact ion-selective electrodes in artificial sweat and achieved root-mean-squared error of 1.37, 1.44, 1.78, 2 mV for Na^+ , K^+ , Li^+ , Pb^{2+} ions^{R17}. Despite the insufficient reference on multi-ion sensing accuracy in mixture, the testing solutions, the targeting ions, as well as the definition of accuracy for different works were not standardized. Therefore, it is difficult to have a direct comparison on accuracies across ion sensors reported in the literature. Fig. 4g demonstrate the importance of having redundant sensors in order to achieve high accuracy. We have edited main text to highlight the number of devices instead of accuracy of 1/8 chip as: “*Compared to the accuracies stemming from 25 devices (1/8 chip), which already contains more devices than most works on ion sensors (Supplementary Table S1), we show that large scale integration, with over 200 devices, can improve the accuracies more than 20 percentage points.*”

9. The author performed all the measurement on artificial bio-fluids. Did the authors perform the measurement on real bio-fluids, and which are the main possible challenges related to this? Is the applied technology and its underlying methods sufficiently robust?

Our response:

We performed the *I-V* characterization of the as-fabricated graphene sensors in human serum and the result is shown in Fig. R14. The sensing chip is still fully functional in real bio-fluids with a high yield of 239/256 devices. A significant shift of the *I-V* characteristics is observed from the baseline measured in water, due to the change in ionic strength and the presence of proteins and steroids. This result shows that the sensing chip is robust enough to function in real bio-fluid and to have the same reliable measurements and potentially high functionality as we have demonstrated in the main paper. We have added the Fig. R14 as Fig. 4f in the main text.

In real-world bio samples, not only will the electrolyte profile be more complex, but there are also proteins such as antibodies, antigens, and hormones that may bring in more confounders in the analysis. Current level, cross-sensitivity and detection limit will be different due to the interference from the complex environment. For a traditional sensor where only one type of sensor is present and absolute selectivity is assumed, the sensor readings with real-world biological samples will not be accurate. Our multiplexed sensing system however analyzes the collective response of array instead of assuming absolute sensitivity or selectivity of one device. It is possible

for the sensor system to be calibrated and trained with sets of real biological samples. Models trained with such data can be very practical and targeted towards specific disease. However, the collection of real biological sample is outside of the scope of this paper. The goal of this paper is to demonstrate a novel approach to tackle the large variations and uncertainties in advanced materials-based sensors as well as bring insights the collection and analysis of biomedical data

We have added the following discussion in the main text to address this comment:

“In additional to complex electrolyte profiles, biological fluids also contain antibodies, antigens, and hormones that introduce more confounders into the analysis. The graphene-based sensors will behave differently in terms of current level, Dirac Point, cross-sensitivity, detection limits as so on (Fig. 4f). The multiplexed sensor array analyzes the collective response of an array instead of assuming absolute sensitivity or selectivity of one device. Models can be readily calibrated or re-trained using real-world biological samples in future studies.”

Figure R14. I - V characteristics of graphene sensor array tested in water (yellow curves) and in human serum (blue curves).

10. From the manuscript it seems that the authors conducted all measurements on a single chip. There is no suggestion in the manuscript that more than one chip was reproduced. How is the reproducibility?

Our response:

We would like to thank the reviewer for the question. There were multiple chips fabricated and tested to make sure the results were reproducible. Shown in Fig. R15 is the sensitivity of two sets of chips with different ISMs functionalization. The average and distribution of the ion sensitivities are very similar from chip to chip. We have added the following sentence in the main text to highlight this *“All characterizations of ISM functionalized sensors were completed at least*

twice with different batches of sensor chips with results remaining consistent (Supplementary Fig. 9)”

Figure R15. Comparison of sensitivity distribution of two batches of sensing chips with different ISMs functionalization. Similar sensor behavior and sensitivity distribution is observed across different sensing chips.

11. The author show several I_{sd} - V_{gs} characteristics. Have the author conducted any characterization of the I_{sd} - V_{sd} for all the sensors? How large is the resistance range of all the sensors after fabrication (before applying liquid-phase environment)?

Our response:

We appreciate the reviewer’s comment. We have performed I_{DS} - V_{DS} characterization of the devices after fabrication and extracted the channel resistance as shown in Fig. R16. The sensor chip is measured in water with $V_{GS} = 0$ V and V_{DS} sweep from 0 to 0.5 V. Shown in Fig. R16b is the extracted channel resistance at different V_{GS} bias. The range of channel resistance varies with different biasing condition. We have added the following figures in supplementary information as Supplementary Fig. 3b and Supplementary Fig. 3d and mentioned them in the main text as

“The drain current with respect to gate voltage (will be referred as “I-V”) characteristics as well as the drain current with respect to drain voltage characteristics of the sensor chip before functionalization are shown in Supplementary Fig. 3a-b...Large device-to-device variations in terms of current level, channel resistance, Dirac Point, (the location of the minimum conduction point), and shape of the I-V characteristics are present for the >200 working devices on one sensing chip (Supplementary Fig. 3)”

Figure R16. a I_{DS} - V_{DS} characteristics of an as-fabricated graphene sensing chip without functionalization. Sensor chip is tested in water with zero gate bias. **d** graphene channel resistance distribution under various gating condition

The following minor typographic corrections need to be corrected:

1. On page 5, in the sentence "The quality of the intrinsic graphene film is analyzed by Raman Spectroscopy as shown in Supplementary Fig. 1b-c", obviously, Supplementary Fig. 1b is not the Raman characterization result, it should be "The quality of the intrinsic graphene film is analyzed by Raman Spectroscopy as shown in Supplementary Fig. 1c-d".
2. Page 13 : "The score of PC1 and PC2 is plotted in Fig. 3e. ". It should be "Fig. 3d".
3. Page 17 - typo, "The graphene film was single-layered with minimal defects as shown in the Raman spectrum in Supplementary Fig.1b-c", it should be "Fig.1c-d"
4. Caption of Supplementary Figure 1 (c), "c Raman spectrum the intrinsic graphene sheet before any fabrication, here "of" is missing between "spectrum" and "the".

Our response:

We would like to thank the reviewer and we revised the manuscript accordingly.

Reply to reviewer 4's remarks:

It is a very well written paper but real time monitoring of real sample can't be demonstrated by using simply urine samples... Two questions: sensing in plasma fluids are in presence of a high

(9g/L) of NaCl concentration and in presence of complex mixture of proteins. The high concentration of salt should decrease the sensitivity of FET sensing scheme, how do the authors aim to overcome this? Same question in a complex fluid with grams per millilitre of serum albumin, sugar, etc.

Our response:

Thank you for this comment. Indeed, the solution environment for real-world bio samples are more complex with high ion concentration and complex proteins including albumins, antibodies, antigens, hormone that may bring in more confounders in the analysis. Shown in Fig. R3a below is the as-fabricated sensor array measured in water and in human serum. A significant shift of the I - V characteristics is observed due to change in ionic strength and the presence of proteins and steroids. In addition, current level, cross-sensitivity and detection limit will also be different due to the interference from the complex environment. For a traditional sensor where only one type of sensor is present and absolute selectivity is assumed, the sensor readings with real-world biological samples might not be accurate. Our sensing system however introduced new approaches to tackle this problem, where a multiplexed sensor array was developed and combined with statistical/machine learning methods. We have demonstrated the robustness and effectiveness of the system in the main text as it is 1) reproducible and reliable by averaging out noise from device-to-device variation; 2) stable and functional in complexed environment and 3) generate multi-dimensional dataset that is compatible with machine learning algorithm to perform complicated tasks.

In order to have a practical functionality in clinical studies, the sensor system would need to be calibrated and trained with large sets of real biological samples. Models trained with such data can be very practical and targeted towards specific disease. Such experiments are possible and will be included in future work, but the collection of real biological samples is outside of the scope of this paper, and it requires other advanced techniques to acquire precise ground truth labels. Instead, we trained another model to simulate how the system would be used in a real-world setting, where biofluids are analyzed using the concentration profile of the K^+ , Na^+ Ca^{2+} ions as biomarkers for electrolyte imbalance disorders. Here we re-categorized the mixture solution sets as shown in the Table R1 below and trained the model to predict the mixture concentration profile instead of the concentration of individual ions. We categorized the mixture solution sets following the relevant ion concentration in human plasma into four classes: Higher Calcium, Higher Potassium, Lower Sodium and Baseline. Diseases correlated to higher calcium (hypercalcemia) in blood include primary hyperparathyroidism, malignancy, thiazide diuretics, kidney diseases and so on^{R3}. Possible causes for higher potassium (hyperkalemia) are chronic kidney diseases, diabetes, congestive heart failure, Addison's diseases etc^{R1}. Lower sodium (hyponatremia) in blood is more common in older people due to the higher likelihood in medication and hospitalization. Medications like diuretics, antidepressants, and carbamazepine (anti-seizure) are also risk factors for hyponatremia^{R2}.

The confusion matrix of the result model is shown in Fig. R3b below. We achieved an average accuracy of 84.7% for classification of ion concentration profiles with solutions containing multiple ion types. The model is less confident at identifying higher calcium and higher potassium. It could be due to the imbalanced training data in such classes, i.e., there are fewer observations with a higher calcium/potassium level that make the learning harder. This analysis demonstrates the possibility of coupling the multiplexed sensing array with machine learning models to achieve fast and high accuracy of electrolyte imbalance-related disease diagnosis. The performance of the

model could be greatly improved with a larger, balanced training dataset. The model could also be easily re-trained towards specific diseases with training data collected with real biofluid from patients in a clinical trial.

To address the reviewer’s comment, we add the following discussion in the main text and SI:

“In additional to complex electrolyte profiles, biological fluids also contain antibodies, antigens, and hormones that introduce more confounders into the analysis. The graphene-based sensors will behave differently in terms of current level, Dirac Point, cross-sensitivity, detection limits as so on (Fig. 4f). The multiplexed sensor array analyzes the collective response of an array instead of assuming absolute sensitivity or selectivity of one device. Models can be readily calibrated or re-trained using real-world biological samples in future studies. Here we simulate how the system would be used in a real-world setting, where biofluids are analyzed using the concentration profile of the K^+ , Na^+ Ca^{2+} ions as biomarkers for electrolyte imbalance disorders. We re-categorized the mixture solution and trained the model to predict the mixture concentration profile instead of the concentration of individual ions (Supplementary Table 4). The confusion matrix of the result model is shown in Supplementary Fig. 24. We achieved an average accuracy of 84.7% for classification of ion concentration profiles with solutions containing multiple ion types. The model is less confident at identifying higher calcium and higher potassium. This could be due to the imbalanced training data in these classes, i.e., there are fewer observations with a higher calcium/potassium level that make the learning more difficult. This analysis demonstrates the possibility of coupling the multiplexed sensing array with machine learning models to achieve fast and high accuracy electrolyte imbalance-related disease diagnoses. The performance of the model could be greatly improved with a larger, balanced training dataset. The model could also be readily re-trained towards specific diseases using training data collected with real biofluid from patients in a clinical trial.”

“Random Forest algorithm was used to demonstrate enhanced functionality of accurate ion type classification, concentration prediction, and potential applications in electrolyte imbalance-related disease diagnostics”

b

	classification			
Lower Sodium	0.941	0.019	0.041	0.000
Baseline	0.003	0.893	0.095	0.009
Higher Calcium	0.035	0.066	0.728	0.171
Higher Potassium	0.004	0.039	0.129	0.828
	Higher Calcium	Baseline	Higher Calcium	Higher Potassium

Figure R3: **a** I - V characteristics of graphene sensor array tested in water (yellow curves) and in human serum (blue curves). **b** confusion matrix for electrolyte imbalance classification in mixture solutions using integrated graphene sensing chip.

Table R1. Solution concentration of mixture solutions and label used for electrolyte imbalance classification model

Solution name	Ca ion concentration	K ion concentration	Na ion concentration	Label
Mix_Ca_1mM	1mM	5mM	30mM	Baseline
Mix_Ca_10mM	10mM	5mM	30mM	Higher Calcium
Mix_Ca_100mM	100mM	5mM	30mM	Higher Calcium
Mix_K_1mM	1mM	1mM	30mM	Baseline
Mix_K_10mM	1mM	10mM	30mM	Higher Potassium
Mix_K_100mM	1mM	100mM	30mM	Higher Potassium
Mix_Na_100μM	1mM	5mM	100μM	Lower Sodium
Mix_Na_1mM	1mM	5mM	1mM	Lower Sodium
Mix_Na_10mM	1mM	5mM	10mM	Lower Sodium
Mix_Na_100mM	1mM	5mM	100mM	Baseline

Reference:

1. Viera, A. J. & Wouk, N. Potassium Disorders: Hypokalemia and Hyperkalemia. **92**, (2015).
2. Reynolds, R. M., Padfield, P. L. & Seckl, J. R. Disorders of sodium balance. *British Medical Journal* vol. 332 702–705 (2006).
3. Nyein, H. Y. Y. *et al.* A Wearable Electrochemical Platform for Noninvasive Simultaneous Monitoring of Ca²⁺ and pH. *ACS Nano* **10**, 7216–7224 (2016).
4. Wang, H., Wu, Y., Cong, C., Shang, J. & Yu, T. Hysteresis of electronic transport in graphene transistors. *ACS Nano* **4**, 7221–7228 (2010).
5. Jung, S.-H. *et al.* Super-Nernstian pH Sensor Based on Anomalous Charge Transfer Doping of Defect-Engineered Graphene. *Cite This Nano Lett* **21**, 42 (2021).
6. Knopfmacher, O. *et al.* Nernst limit in dual-gated Si-nanowire FET sensors. *Nano Lett.* **10**, 2268–2274 (2010).
7. Lin, P., Yan, F. & Chan, H. L. W. Ion-sensitive properties of organic electrochemical transistors. *ACS Appl. Mater. Interfaces* **2**, 1637–1641 (2010).
8. Zhang, M. *et al.* High-performance dopamine sensors based on whole-graphene solution-gated transistors. *Adv. Funct. Mater.* **24**, 978–985 (2014).
9. Goodwin, D. G. *et al.* Faradaic effects in electrochemically gated graphene sensors in the presence of redox active molecules. *Nanotechnology* **31**, 405201 (2020).
10. Rudnitskaya, A. Calibration update and drift correction for electronic noses and tongues. *Frontiers in Chemistry* vol. 6 (2018).
11. Fu, W. *et al.* Graphene transistors are insensitive to pH changes in solution. *Nano Lett.* **11**, 3597–3600 (2011).
12. Ferrari, A. C. *et al.* Raman spectrum of graphene and graphene layers. *Phys. Rev. Lett.* **97**, 187401 (2006).
13. Ferrari, A. C. & Basko, D. M. Raman spectroscopy as a versatile tool for studying the properties of graphene. *Nat. Nanotechnol.* 2013 **84** **8**, 235–246 (2013).
14. Loon, J. D., Groenen, L. C., Verboom, W., Reinhoudt, D. N. & Wijnemga, S. S. Upper Rim Calixcrowns: Elucidation of the Mechanism of Conformational Interconversion of Calix[4]arenes by Quantitative 2-D EXSY NMR Spectroscopy. *J. Am. Chem. Soc.* **113**, 2378–2384 (1991).
15. Eroglu, E. *et al.* Nitrate uptake by p-phosphonic acid calix[8]arene stabilized graphene. *Chem. Commun.* **49**, 8172–8174 (2013).
16. Fakih, I. *et al.* Selective ion sensing with high resolution large area graphene field effect transistor arrays. *Nat. Commun.* **11**, 1–12 (2020).
17. Hanitra, I. N., Criscuolo, F., Carrara, S. & De Micheli, G. Multi-Ion-Sensing Emulator and Multivariate Calibration Optimization by Machine Learning Models. *IEEE Access* **9**, 46821–46836 (2021).

REVIEWERS' COMMENTS

Reviewer #1 (Remarks to the Author):

The authors give reasonable explanations for the submitted questions. The revised paper is suitable to be published in Nature Communications. However, there are still some suggestions on how to improve the paper before publication. These suggestions are given based on the authors' point-by-point responses.

Question 3. Since there are more than 200 devices on the same sensor, the multi-channel measurement is important for the data collection. How many devices could be measured simultaneously in the testing process? How long does it take to finish the measurements on all the 215 devices in a test?

Our response:

We thank the reviewer for the question. The custom PCB measurement system is able to measure all sensors almost simultaneously. This is achieved by the 16-channel multiplexer on the board to switch quickly between different rows of the sensor array. For a standard VGS sweep from -0.6V to 0.9V with 20mV/s sweeping speed, it takes 1.5 mins to finish the VGS sweep for one sensor cell. With the current multiplexer, it roughly takes 3 minutes to finish the I-V measurement of all 256 devices. We have added the following sentence in the main text to reflect reviewer's comment: "The change in the source-drain current I_{DS} for each row and column combination of the sensor array are automatically measured as a function of gate-source and drain-source voltages, VGS and VDS. For a standard VGS sweep (from -0.6V to 0.9V with 20mV/s sweeping speed), it takes roughly 3 minutes to finish measuring

New comment/suggestion:

The authors' explanation is reasonable, but more details about the measurements could be given in the supplementary information. According to Figure 1c and Figure S1e, the switch between different sensor units is based on the 16-channel multiplexer for row/column selection on the 16 × 16 sensor array. The high-throughput measurement is impressive since it only takes 3 minutes to finish all the measurements (1.5 minutes for a single cell). Herein, it's essential to clarify how to realize the measurements on different columns/rows simultaneously.

Question 5. In Algorithm-enhanced sensing accuracy section, the authors should clarify if the multi-ion solutions used in the second set of experiments were in water or AU/AEP. It would be better if the authors could provide the sensing performance and classification results with real-world biological samples to support the practical significance of the sensing platform. Or the authors should at least discuss on how the sensing performance and data analysis could be affected due to the complexity of real-world biological samples.

Our response:

We appreciate the reviewer's comment. The multi-ion solution used in the model training was in water. We have clarified in the main text as: "The second set, called "mixture solutions", contains electrolyte solutions with multiple ion types in deionized water and will be discussed in the following section."...

...

..."Random Forest algorithm was used to demonstrate enhanced functionality of accurate ion type classification, concentration prediction, and potential applications in electrolyte imbalance-related disease diagnostics".

New comment/suggestion:

We appreciate the explanations from the authors. Firstly, the labels in Table R1 are not consistent with Table S4, and we think there may be some errors in Table S4. For example, the "Mix_K_1mM" should be "baseline", but not "higher potassium" in Table S4 (The label of "Mix_K_1mM" is "baseline" in Table R1, which should be correct). These errors should be corrected in the final draft.

Secondly, we notice that the results in Figure S24 show that the classification of higher calcium

solution (which is the classification of solution with at least 10 mM of Ca²⁺ and solution with only 1 mM of Ca²⁺ in the mixture) is less confident, which is consistent with the conclusion in Figure 4e. We suggest that the authors could discuss common ground between these two in the manuscript.

6. For the PCA, how many features were applied for the classification? The authors should clarify these features in the manuscript.

Our response:

We appreciate the reviewer's question. The number of features (dimensions) equals to the number of working devices on the sensing chip. PCA was used to reduce the dimensionality from >200 to 3 principal components. We performed two separate PCA analysis with

1) Dirac point as the feature value: The PCA matrix e has a dimension of $N \times M$ where N is the pure solution set with $N = \{K+100\mu M, K+1mM, K+10M, K+100mM, Na+100\mu M, Na+1mM, Na+10M, Na+100mM, Ca_2+100\mu M, Ca_2+1mM, Ca_2+10M, Ca_2+100mM\}$. The M features are the Dirac Point of individual devices on the sensing chip measured in each pure solution. The first and second principal component is plotted in Fig. R4a. The three types of ions are well separated with a total variance of 92.1%. This analysis demonstrates the multi-variate nature of the dataset produced by the highly integrated sensing chip with 3 ISMs.

2) Sensitivity towards K⁺, Na⁺, Ca²⁺ solutions of each individual device as features: The sensitivity is extrapolated as the slope of change in Dirac Point with respect to different concentrations. The PCA matrix has a dimension of $3 \times M$. The PCA is plotted in Fig. R4b and the first two principal components cover 100% of the total variance in the dataset. This result demonstrated that the sensitivity profile of the integrated sensing chip is significantly different and can be used to fingerprint ion types.

We modified the main text and added a paragraph explaining the details of PCA in the methods as: "We first performed Principal Component Analysis (PCA) using the Dirac Point of individual devices as features to visualize the multivariate data under a lower dimensional space while preserving the largest variance. The first two principal components (PC) accounted for 92.1% of the total variance in the data. The score of PC1 and PC2 is plotted in Fig. 3e and the details of the PCA analysis are explained in Supplementary Note 7. The clusters of K⁺, Na⁺ and Ca²⁺ ions are well-separated, indicating the sensor's ability to distinguish between different types of ions in electrolyte. Further separation can be achieved by using the ion sensitivities of individual sensors as the feature set as shown in Supplementary Fig. 20."

New comment/suggestion:

We appreciate the explanations from the authors. According to the authors' response and Figure R4a (Figure 3d), $N = 12$ in the dataset for model training, which is much smaller than M (the number of features). However, the number of samples is commonly larger than the number of features in PCA and other models such as LDA. We suggest the authors give more details about the realization of PCA in the supplementary information. To make N larger than M , there may be multiple replicates of each pure solution. The number of replicates of each solution could be included in the manuscript. (We have noticed the comparable results on two different chips, which are in the answer to review 3's question 10).

Reviewer #2 (Remarks to the Author):

I am pleased to see that the authors have improved the current version of the manuscript substantially through additional experiments and discussions. The findings are more robust than in the original version. I feel that the point-to-point replies to reviewers' comments are satisfactory. I recommend it to be published in Nature Communications.

Reviewer #3 (Remarks to the Author):

In this resubmitted manuscript, Xue et al. have considered the majority of the major concerns raised in my previous report and those of the other reviewers. Therefore, I am pleased to state that the manuscript is ready for publication, once the following minor comment has been considered:

. On page 14, the sentence "The score of PC1 and PC2 is plotted in Fig. 3e. and ...". In the main text, there is no "Fig. 3e", it should be "Fig. 3d".

Reviewer #4 (Remarks to the Author):

The authors have addressed my comments in a satisfactory manner.

Reply to reviewer 1's remarks:

The authors give reasonable explanations for the submitted questions. The revised paper is suitable to be published in Nature Communications.

However, there are still some suggestions on how to improve the paper before publication. These suggestions are given based on the authors' point-by-point responses.

Our response:

We thank the reviewer for the acknowledgment on the novelty and impact of our work. We also thank the reviewer for the constructive comments, which have significantly helped to increase the clarity of our manuscript.

Question 3. Since there are more than 200 devices on the same sensor, the multi-channel measurement is important for the data collection. How many devices could be measured simultaneously in the testing process? How long does it take to finish the measurements on all the 215 devices in a test?

New comment/suggestion:

The authors' explanation is reasonable, but more details about the measurements could be given in the supplementary information. According to Figure 1c and Figure S1e, the switch between different sensor units is based on the 16-channel multiplexer for row/column selection on the 16×16 sensor array. The high-throughput measurement is impressive since it only takes 3 minutes to finish all the measurements (1.5 minutes for a single cell). Herein, it's essential to clarify how to realize the measurements on different columns/rows simultaneously.

Our response: We thank the reviewer for recognizing the data acquisition power of our platform. The custom PCB measurement system leverages very high-speed electronics to rapidly scan the sensor array such that measurements are virtually simultaneous relative to the time scales at which these chemical sensors respond. The scan rate for the sensor array is **~10** frame per second whereas the sensor response is **5-7** seconds as shown in Supplementary Figure 12. This is achieved using low-noise and high-speed transimpedance amplifiers to accurately amplify the small sensor currents. High-speed 16-channel analog multiplexers with low on-resistance (2.5 ohms) are then used to very quickly switch between different rows and columns of the sensor array. We have included a more detailed description of our measurement system in the Methods section as shown below:

The measurement system makes use of an Atmel SAM3X microcontroller with an 84 MHz clock, which enables very high-speed data acquisition. Dual 12-bit digital-to-analog converters (DACs) are employed to vary the V_{DS} and V_{GS} voltages applied throughout the sensor array appropriately depending on the measurement configuration (e.g. I-V sweep, transient I_{DS}). The microcontroller is paired with a custom printed circuit board (PCB) designed to precisely match the input and output ports of the microcontroller in order to achieve an overall compact form factor and portable measurement system (about the size of a cell phone). The custom PCB includes 16 transimpedance amplifiers (one for each column of the sensor array) that make use of 2-stages along with low-noise resistors and operational amplifiers to achieve an overall gain of 10,000 V/I so that μA (and sub- μA) sensor currents can be amplified to the appropriate voltage range and measured with very high accuracy using a 12-bit analog-to-digital converter (ADC).

Row and column selection is performed using 16x1 bidirectional analog multiplexers with low on-resistance (2.5Ω) to minimize distortion of the applied V_{DS} voltages and sensor current readouts. One analog multiplexer is used to apply V_{DS} along a single row of devices. The resulting column currents are all continuously amplified using the 16 transimpedance amplifiers. The second analog multiplexer is used to rapidly switch which transimpedance amplifier output is applied to the ADC for readout. After all column currents have been read out, V_{DS} can be applied to the subsequent row and the process is repeated. In this way, we are able to rapidly scan the entire array of devices. Low dropout regulators are employed so that the entire measurement system can be conveniently powered using a single universal serial bus (USB) power supply. All measurement instructions and results are transmitted via USB as well.

Question 5. In Algorithm-enhanced sensing accuracy section, the authors should clarify if the multi-ion

solutions used in the second set of experiments were in water or AU/AEP. It would be better if the authors could provide the sensing performance and classification results with real-world biological samples to support the practical significance of the sensing platform. Or the authors should at least discuss on how the sensing performance and data analysis could be affected due to the complexity of real-world biological samples.

New comment/suggestion:

We appreciate the explanations from the authors. Firstly, the labels in Table R1 are not consistent with Table S4, and we think there may be some errors in Table S4. For example, the “Mix_K_1mM” should be “baseline”, but not “higher potassium” in Table S4 (The label of “Mix_K_1mM” is “baseline” in Table R1, which should be correct). These errors should be corrected in the final draft.

Secondly, we notice that the results in Figure S24 show that the classification of higher calcium solution (which is the classification of solution with at least 10 mM of Ca^{2+} and solution with only 1 mM of Ca^{2+} in the mixture) is less confident, which is consistent with the conclusion in Figure 4e. We suggest that the authors could discuss common ground between these two in the manuscript.

Our response:

We thank the reviewer for this constructive comment and suggestion. We have corrected the labels in Supplementary Table 4 as shown below and added the following discussion in the main text to address the models' confidence level of higher Ca^{2+} solutions:

The lower confidence on concentrated Ca^{2+} solutions is also observed in the Ca^{2+} ion concentration model shown in Fig. 4e. A possible reason could be the intrinsic lower Nernstian slope for bivalent ion and the choice of features. Quantifying higher calcium concentrations could be further improved by carefully redesigning the functionalization matrix and optimizing feature selection.

Supplementary Table 4. Mixture solution set ion concentration for sensor testing

Solution name	Ca^{2+} ion concentration	K^{+} ion concentration	Na^{+} ion concentration	Label
Mix_Ca_100 μM	100μM	5mM	30mM	-
Mix_Ca_1mM	1mM	5mM	30mM	Baseline
Mix_Ca_10mM	10mM	5mM	30mM	Higher Calcium
Mix_Ca_100mM	100mM	5mM	30mM	Higher Calcium
Mix_K_100 μM	1mM	100μM	30mM	-
Mix_K_1mM	1mM	1mM	30mM	Baseline
Mix_K_10mM	1mM	10mM	30mM	Higher Potassium
Mix_K_100mM	1mM	100mM	30mM	Higher Potassium
Mix_Na_100 μM	1mM	5mM	100μM	Lower Sodium
Mix_Na_1mM	1mM	5mM	1mM	Lower Sodium
Mix_Na_10mM	1mM	5mM	10mM	Lower Sodium
Mix_Na_100mM	1mM	5mM	100mM	Baseline

6. For the PCA, how many features were applied for the classification? The authors should clarify these features in the manuscript.

New comment/suggestion:

We appreciate the explanations from the authors. According to the authors' response and Figure R4a (Figure 3d), $N = 12$ in the dataset for model training, which is much smaller than M (the number of features). However, the number of samples is commonly larger than the number of features in PCA and other models such as LDA. We suggest the authors give more details about the realization of PCA in the supplementary information. To make N larger than M , there may be multiple replicates of each pure solution. The number of replicates of each solution could be included in the manuscript. (We have noticed the comparable results on two different chips, which are in the answer to review 3's question 10).

Our response:

We thank the reviewer for the comment. Contemporary dataset for PCA does have $N > M$, where N is the number of observations and M is the dimensionality (features). It is also possible to perform PCA with consistent result with high dimensionality dataset, where $N \ll M$. The steps we took to perform PCA is listed below:

1. Centered the columns of the PCA matrix X (with a dimension of $N \times M$, $N \ll M$) by subtracting column mean
2. Performed Singular value decomposition (SVD) on matrix: $X = USV^T$
 - a. U : left singular matrix
 - b. S : diagonal matrix of singular values corresponding to the estimated principal component
 - c. V : right singular vector corresponding to the estimated principal directions
3. Calculate the score of the first two principal components by $C = SV$
4. Plot the first two principal components

We have included this information in Supplementary Information Note. 7. There are ways to increase N/M ratio, such as including multiple measurements (as the reviewer suggested), random sampling the devices and so on. Here we would like to showcase and visualize the variance of the raw data generated by one highly integrated sensing chip thus chose to perform the PCA as explained above.

Reply to reviewer 2's remarks:

I am pleased to see that the authors have improved the current version of the manuscript substantially through additional experiments and discussions. The findings are more robust than in the original version. I feel that the point-to-point replies to reviewers' comments are satisfactory. I recommend it to be published in Nature Communications.

Our response:

We thank the reviewer for all the constructive feedback and suggestions, which have significantly helped to increase the clarity of our manuscript. We appreciate the reviewer's supportive consideration for its publication on *Nature Communications*.

Reply to reviewer 3's remarks:

In this resubmitted manuscript, Xue et al. have considered the majority of the major concerns raised in my previous report and those of the other reviewers. Therefore, I am pleased to state that the manuscript is ready for publication, once the following minor comment has been considered:

. On page 14, the sentence "The score of PC1 and PC2 is plotted in Fig. 3e. and ...". In the main text, there is no "Fig. 3e", it should be "Fig. 3d".

Our response:

We thank the reviewer for positive comment on the resubmitted manuscript and support for its publication on *Nature Communications*. We also appreciate the reviewer for the constructive comments and suggestions throughout the revision process. We have edited the manuscript accordingly.

Reply to reviewer 4's remarks:

The authors have addressed my comments in a satisfactory manner.

Our response:

We thank the reviewer for the acknowledgment on the quality of our work. We appreciate the reviewer's supportive consideration for its publication on *Nature Communications*.